



# High-resolution modelling of long-term trends in the oxygen and carbon cycles of the Benguela upwelling system

Katharina D. Six[1] and Uwe Mikolajewicz[1]

[1]Max Planck Institute for Meteorology, Hamburg, Germany

**Correspondence:** Katharina Six (katharina.six@mpimet.mpg.de)

**Abstract.** We investigate driving forces of the biogeochemistry of the Benguela upwelling system (BUS) and their temporal changes over the 20th century. For this purpose, we developed a global ocean-only model in a stretched grid configuration, which resolves meso-scale circulation structures in the area of interest. The biogeochemical module of this model is extended by a more comprehensive nitrogen cycle to account for the specific nitrogen loss processes common in eastern upwelling systems. The model is forced by 110 years of atmospheric reanalysis data. To assess the impact of meso-scale circulation structures on local biogeochemical processes we compare our results to a set-up with a coarser horizontal resolution, comparable to the ocean component of Earth system models used for anthropogenic climate projections. In the higher spatial resolution we find enhanced intermediate depth ventilation (200-1000 m) with concurrent reduced loss of bioavailable nitrogen and a high shelf-bound biological production, in line with observations. Moreover, only in the high resolution setup do multi-decadal trends of deoxygenation match observation-based estimates. Our study supports the view that the presence of meso-scale circulation structures exerts a major influence on biogeochemical patterns, especially on mid-depth oxygen concentrations. Furthermore, we show for the first time that by including this high spatio-temporal variability of the circulation, the regional anthropogenic carbon uptake of the BUS over the 20th century is lower than in the coarse resolution model. This indicates that, at least for some regions, the pathway of changes in the marine biogeochemistry as projected by state-of-the-art coarse resolution Earth system models is associated with high uncertainty.





## 1   Introduction

There is increasing interest in the role of coastal upwelling regions within the climate system of the Earth (Gruber et al., 2012; Emeis et al., 2018). These regions, and especially the Benguela upwelling system (BUS), host the most productive
ecosystems with corresponding high organic carbon fluxes to the ocean interior. Decay of organic material in combination with the hydrodynamical characteristic of a low ventilation of intermediate depths (200 to 1000 m) creates the world's largest oxygen minimum zones (OMZs) (Cabré et al., 2015). Furthermore, high organic fluxes, often referred to as biological carbon pump, also contribute to the global marine carbon storage. Climate induced changes in coastal upwelling systems put multiple stresses on their ecosystems which might cause unprecedented impacts on the global carbon cycle (Gruber et al., 2012; Bopp
et al., 2013). It is hypothesised that ocean warming increases deoxygenation (Schmidtko et al., 2017) which could affect nutrient cycling and the biological pump with potentially adverse consequences on marine ecosystems. Increasing levels of atmospheric $CO_2$ lead to a lowering of seawater pH. Having already naturally low levels of seawater pH, the biological ecosystem in coastal upwelling regions are particularly vulnerable to ocean acidification (Gruber et al., 2012). Thus, identifying controls of biogeochemical processes in coastal upwelling systems is essential to assess the potential impacts on deoxygenation and
regional carbon budgets in a changing climate.

A common feature of eastern boundary upwelling systems, including the BUS, is their high spatial and temporal variability in physical processes in response to coastal jets and a quasi-permanent wind-induced upwelling modulated by a wind stress curl driven current system (Fennel, 1998). Assessing mean state and temporal evolution of the biogeochemistry in coastal upwelling systems is, therefore, an ambitious task. The high spatio-temporal variability of the physical drivers challenges observational
methods to achieve a comprehensive picture. Modelling approaches could give insights in the physical and biogeochemical pattern of the BUS. However, none of the global Earth System models, which are currently used for future climate projections, is capable to reproduce present-day oxygen concentrations in the BUS (Séférian et al., 2020) or past decadal trends (Buchanan and Tagliabue, 2021). They overestimate the size of the OMZ in the eastern South Atlantic near the coast of South Africa. The simulated multi-model mean of the decadal $O_2$ trend is far too weak because models show no consistency in the trend (see
Fig. S1 in Buchanan and Tagliabue (2021)). Séférian et al. (2020) postulated a systematic bias in the ocean biogeochemical models which seems to be independent from ocean resolution or complexity of marine biogeochemical models. However, the horizontal nominal resolution of the considered Earth system models is typically around 100 km; only one could be classified as eddy-permitting (25 km). Therefore, it is debatable if the simulated bias in the OMZs is not attributable to the low spatial model resolutions which do not resolve meso-scale structures. These structures, being eddies and filaments, are known to in-
troduce strong ocean mixing and water column ventilation. Recently, a global eddy resolving coupled ocean model (spatial resolution of 0.1 degrees, Frenger et al. (2018) shows a more realistic oxygen distribution and emphasizes the importance of meso-scale eddies for the representation of biogeochemical processes. Regional modeling approaches typically resolve these meso-scale structures and simulate more realistic OMZs (Gutknecht et al., 2013; Schmidt and Eggert, 2016; Mohrholz et al., 2008). However, to capture long-term trends regional models have to prescribed realistic time-dependent and consistent bound-
ary conditions for ocean physic and biogeochemistry (Schmidt and Eggert, 2016), which are not available yet. Furthermore,



Espinoza-Morriberón et al. (2021) showed that by suppressing the remote variability, entering the regional model domain at the boundaries, trends of oxygen are strongly weakened or even reversed. In order to avoid these problems, it is advantageous to choose a global model approach with a sufficient spatial resolution to depict meso-scale structures to investigate the mean state and the temporal evolution of oxygen in the BUS.

Another open issue is the low complexity of biogeochemical model in representing processes in suboxic waters ($O_2 < 20$ mmol m$^{-3}$). All ocean components of the global models, analysed in the study by Séférian et al. (2020), include a simplified nitrogen cycle, often accounting only for nitrate ($NO_3^-$), some for ammonia ($NH_4^+$), but none for nitrite ($NO_2^-$). As far as we know, complete denitrification is the only nitrogen loss process in such models. However, as there is no observational data set to evaluate complete denitrification rates, the strength of nitrogen loss and its decadal trends might be biased. Furthermore, the

nitrogen loss process anammox, which is the anaerobic ammonium oxidation with nitrite, is fully ignored despite that there are upwelling systems where anammox is particularly significant (Kalvelage et al., 2013). In addition, depending on the oxygen concentration, nitrite used for anammox could instead be oxidized back to $NO_3^-$. Thus, the fate of nitrite exerts a control on the availability of fixed nitrogen in the ocean (Rixen et al., 2020). Since there are observational data for $NH_4^+$ and $NO_2^-$, these tracers could be used to evaluate the performance of the simulated anaerobic processes in a model with a comprehensive

nitrogen cycle.

Thus, two questions arise: i) does an adequate representation of meso-scale circulation structures have implications for the mean state and the temporal evolution of the oxygen minimum zone in the BUS over the 20th century and, ii) does a comprehensive nitrogen cycle provide more insight in aerobic and anaerobic processes in OMZs ? To the end, we design and apply a new model set-up with a high spatial resolution in the BUS and adjacent regions (higBUS) to represent meso-scale

structures. This physical-biogeochemical ocean model has a global coverage to guarantee hydrodynamical tracer consistency and to capture potential remote impacts of the framing current systems north and south of the BUS. Its design allows for a long spin-up simulation to achieve quasi-steady state of the preindustrial ocean before we start a transient simulation of the 20th century with increasing atmosphere $CO_2$ concentrations. More important, the biogeochemical component, which includes a fully prognostic carbon chemistry and a representation of plankton dynamics, is extended by a newly developed description

of a comprehensive nitrogen cycle ($NO_3^-$, $NH_4^+$, $NO_2^-$) to evaluate anaerobic processes occurring in OMZs. In the following, we will first compare the mean state of ocean physics and biogeochemistry of higBUS to observations. Then, we evaluate simulated decadal oxygen and carbon trends.

## 2    Model environments

### 2.1    General physical ocean model

MPIOM is a state-of-the-art general ocean circulation model based on primitive equations. It is formulated on an Arakawa-C grid in the horizontal and on z-levels in the vertical direction using the hydrostatic and the Boussinesq approximation. Subgrid-scale parameterizations include lateral mixing on isopycnals (Redi, 1982) and the Gent et al. (1995) parametrization for eddy-induced mixing. Vertical mixing is based on a combination of the Richardson-number dependent scheme of Pacanowski and





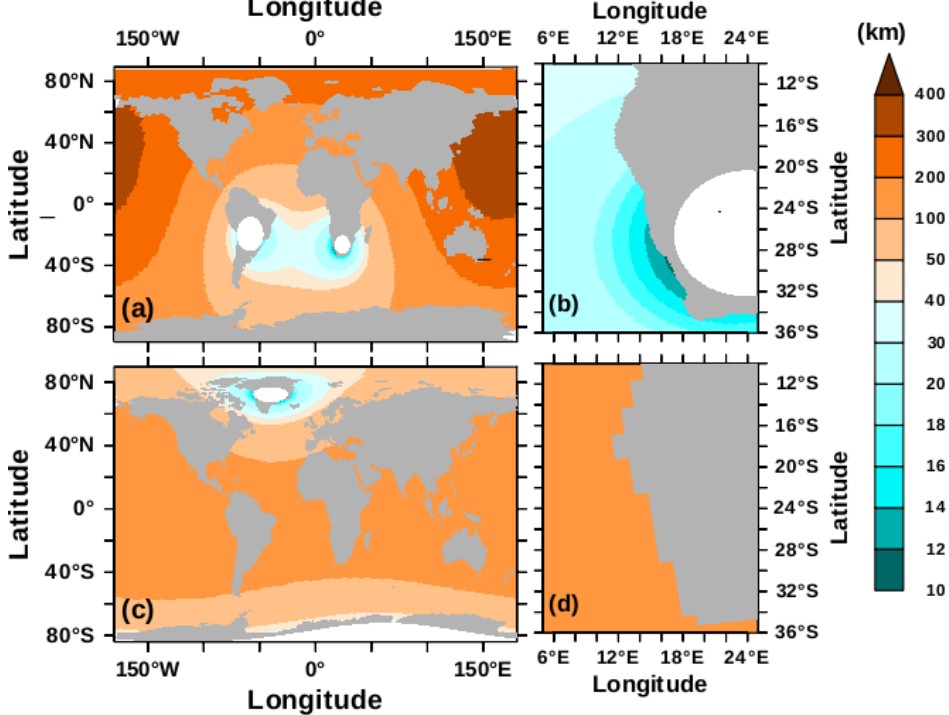

**Figure 1.** Grid resolution in zonal direction of higBUS (a,b) and GR15 (c,d) (km). White circles indicate the location of the grid poles. Panel (b) and (d) show the subregion of 6° E to 25° E and 12° S to 36° S including the Benguela upwelling system.

Philander (1981) and a wind-driven turbulent mixing scheme for the upper ocean. Details on parametrizations of atmospheric
exchange fluxes can be found in Jungclaus et al. (2013).

Here we use MPIOM in two different grid configurations, both covering the entire world ocean. GR15 is the standard ocean model component used within the MPI Earth System Model (Jungclaus et al., 2013; Mauritsen et al., 2019). GR15 is a bi-polar grid with one pole is located on Greenland and one on Antactica. It has a horizontal resolution of 1.5 degrees gradually varying between 15 km near Greenland and more than 180 km in the tropical ocean. The second setup uses a special grid configuration
(higBUS, Fig. 1) where the poles are shifted to South Africa (27° S, 24° E) and South America (20° S, 58° W). Shifting of the pole locations leads to a finer horizontal resolution between 10° S-40° S in the Atlantic (see e.g. Mathis et al. (2019) for the same method applied to the North Atlantic). The Atlantic sector of the Southern Ocean (>60° S) has a similar horizontal resolution in both setups. The location of the poles leads to a fine grid resolution in the area of interest but a relatively coarse resolution of approx. 330 km in the Pacific. The Benguela current system is resolved with approx. 10 km, a resolution which
can be considered as eddy resolving. The Rossby radius of deformation for the BUS is estimated to be about 30 km (Hallberg, 2013; Chelton et al., 1998). In MPIOM, the impact of the parameterization for unresolved eddy-induced transport by Gent et al. (1995) is scaled with the horizontal grid distance, basically turning off this artificial eddy transport in regions such as the BUS in higBUS. The vertical resolution is identical in both setups with 40 layers. In the upper 100 m, layer thicknesses are



around 10 m and they are gradually increasing towards ocean interior (layer thickness max. 600 m). The topography is based
on ETOPO2 (NOAA, 2006) and we consider partial grid cells in the last wet grid cell. The time step is 1 hour in GR15 and 30
min in higBUS. Tidal effects are not included.

## 2.2 Biogeochemical model

One focus of the global biogeochemical model HAMOCC is and has always been on a good representation of large scale
geochemical tracer distributions (Maier-Reimer, 1993). Aspects of the marine biota have been kept as simple as possible. This
version of HAMOCC considers effects of the marine biota by an extended plankton dynamics module representing nutrients
(phosphate, nitrogen species, iron), two types of phytoplankton (bulk and diazotrophs) and one type of zooplankton, detritus
and dissolved organic matter (DOM) (Six et al., 1996; Ilyina et al., 2013; Paulsen et al., 2017). Shell material production
in forms of opal or calcium carbonate is linked to the production of detrius with a preference to opal production if silicate
is available. Organic matter/shell material production and decomposition/dissolution affect the alkalinity. All biogeochemical
tracers are subject to the same hydrodynamics as temperature and salinity. Particulate matter such as detritus and shell material
are additionally subject to gravitational sinking. As long as oxygen is available detritus and DOM are remineralized which
releases carbon and nutrients in a fixed composition ratio ("Redfield ratio", Takahashi et al. (1985)) and consumes oxygen
and alkalinity. In suboxic zones denitrification and sulfate reduction control the decay of detritus. Particles reaching the sea
floor fuel a sediment module with 12 active layers (Heinze et al., 1999). Each sediment layer consists of solid particles and
porewater which interact following the same processes of remineralization, denitrification and sulfate reduction as represented
in the water column. Diffusive exchanges between the sediment layers and the water-sediment boundary supply oxygen and
nitrate to fuel organic matter decay and maintain the reflux of carbon, phosphate, iron, silicate and alkalinity from the sediment
to the water column. Depending on the particle flux and the sediment composition, shells and organic material might be buried
and transferred to deeper sediment layers and eventually end up in the diagenetically consolidated burial layer.

Particle fluxes of organic matter are described by the linearly increasing sinking speed following the formulation of Martin
et al. (1987). In the upper 90 m the sinking speed is kept constant at 3.5 m d$^{-1}$ and it reaches a maximum of 70 m d$^{-1}$ at 6000
m. Shell material is sinking with a constant speed of 30 m d$^{-1}$ for calcareous shells and 22 m d$^{-1}$ for opal frustules. Processes
of shell dissolution and air-sea gas exchange of $CO_2$ are formulated as in Ilyina et al. (2013).

A new development that we are introducing for the HAMOCC version used here is a more comprehensive nitrogen cycle,
which in particular includes a detailed representation of nitrogen loss processes. Nitrate reduction processes occur only in
suboxic to anoxic waters. They potentially lead to the loss of bioavailable nitrogen and, thus, affect directly the biological
production. In the following we concentrate on the description of newly implemented processes of the N-cycle and give only a
general overview of other represented biogeochemical processes.

### 2.2.1 Nutrient cycle

In general, biological activity consumes nutrients in the euphotic surface layer to build organic material. This is partly rem-
ineralized within the upper ocean by zooplankton and bacterial activity and partly exported towards the aphotic deep ocean



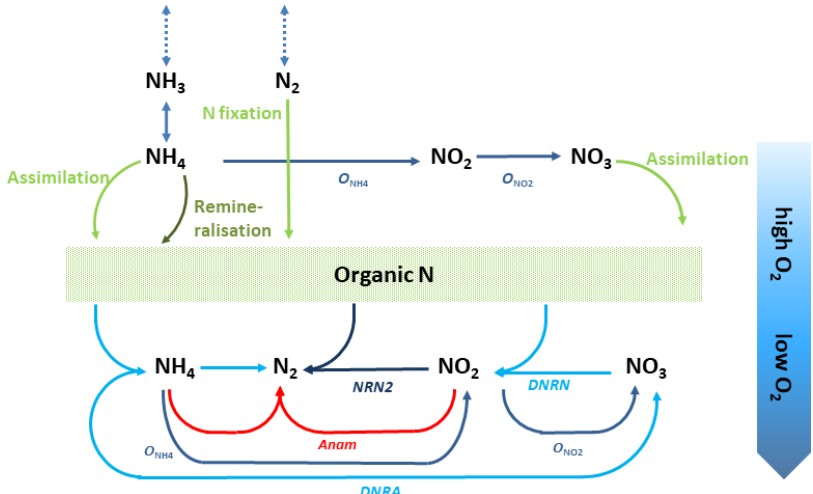

**Figure 2.** Schematic representation of processes included in the comprehensive nitrogen cycle. Solid arrows indicate processes within the water column and, expect for assimilation, the sediment. Dashed arrows represent air-sea exchange. Process labelling is in accordance with Eq.(2)-(15).

and the sediments. The local change in nutrient concentrations occurs due to hydrodynamics, phytoplankton growth, grazing activities, organic matter decay, fluxes to the sediments and weathering input. In case of total inorganic nitrogen (TIN) which comprises nitrate, nitrite, and ammonium we also have to consider air-sea gas exchange for ammonia. In general, all biogenic tracers follow

$$\frac{\partial C^i}{\partial t} + \nabla \boldsymbol{U} C^i - D C^i = \Sigma_k \Psi_k^i(C_k, C^i) + \Psi_{airsea} + \Psi_{sedflux} \tag{1}$$

where $\boldsymbol{U}$ is the three-dimensional advection vector, $D$ the diffusion operator, and $\Psi_k^i(C_k, C^i)$ refers to the source and sink operators for biogeochemical processes, fluxes across the air-sea interface ($\Psi_{airsea}$) and interactions with the sediment ($\Psi_{sedflux}$). In the following paragraphs we describe the processes summarized by $\Psi$ in equation (1) on TIN. A schematic view on the processes is given in Fig. 2. All parameters and rates used in the new N-cycle of the model are given in Table A1 in the Appendix. A detailed description for other prognostic tracers like phosphate, dissolved iron, dissolved inorganic carbon, and alkalinity is found in Ilyina et al. (2013).

$$\Sigma_k \Psi_k(\mathrm{NO_3^-}) \quad := \quad \text{- phytoplankton growth on } \mathrm{NO_3^-} \text{ - dissimilation of } \mathrm{NO_3^-} \text{ to } \mathrm{NO_2^-} \text{ - dissimilation of } \mathrm{NO_3^-} \text{ to } \mathrm{NH_4^+}$$

$$\text{+ oxidation of } \mathrm{NO_2^-} \text{ + anammox}$$

$$\frac{\partial \mathrm{NO_3^-}}{\partial t} \quad = \quad -\sigma_{\mathrm{NO_3}} G_{phy} - DNRN - DNRA$$

$$+ O_{\mathrm{NO_2}} + \epsilon_a Anam \tag{2}$$





Sinks for $NO_3^-$ are phytoplankton growth, and dissimilatory nitrate reduction to nitrite ($DNRN$) or ammonium ($DNRA$). Sources for $NO_3^-$ are oxidation of $NO_2^-$ ($O_{NO_2}$) and as a small byproduct of anammox ($Anam$). Correspondingly, changes of $NO_2^-$ and $NH_4^+$ are given by

$$
\begin{aligned}
\Sigma_k \Psi_k(NO_2^-) \quad :=& \quad \text{dissimilation of } NO_3^- \text{ to } NO_2^- \text{ - denitrification of } NO_2^- \text{ to } N_2 \\
& \quad + \text{ oxidation of } NH_4^+ \text{ - oxidation of } NO_2^- \text{ - anammox} \\
\frac{\partial NO_2^-}{\partial t} =& \quad DNRN - NRN2 \\
& \quad + O_{NH_4} - O_{NO_2} - (1 + \epsilon_a)\, Anam
\end{aligned}
\tag{3}
$$

$$
\begin{aligned}
\Sigma_k \Psi_k(NH_4^+) \quad :=& \quad \text{- phytoplankton growth on } NH_4^+ + \text{remineralization} \\
& \quad + \text{ dissimilation of } NO_3^- \text{ to } NO_2^- + \text{ dissimilation of } NO_3^- \text{ to } NH_4^+ \\
& \quad + \text{denitrification on } NO_2^- \text{ to } N_2 \\
& \quad \text{- oxidation of } NH_4^+ \text{ - anammox + air-sea-gasexchange} \\
\frac{\partial NH_4^+}{\partial t} =& \quad -\sigma_{NH_4} G_{phy} + (F_{pz} + M_{zoo} + \lambda_{dom}\, DOM + F_{det})\, R_{N:P} \\
& \quad + DNRN + DNRA
\end{aligned}
$$

$$
\begin{aligned}
& \quad + NRN2 \\
& \quad - O_{NH_4} - Anam + \Psi_{airsea}
\end{aligned}
\tag{4}
$$

where $NRN2$, the denitrification of $NO_2^-$ to $N_2$, is a total N-loss process of bioavailable nitrogen. $G_{phy}$ is the nutrient consumption due to phytoplankton growth on $NO_3^-$ or $NH_4^+$. The $NH_4^+$ source from remineralization includes decomposition of organic material via remineralization in oxygenated waters ($F_{det}$), grazing activity on phytoplankton $F_{pz}$, grazing activity on

zooplankton $M_{zoo}$, and bacterial decomposition of dissolve organic matter $\lambda_{dom}\, DOM$. In HAMOCC, we use a constant elementary composition of organic material following the "Redfield" ratio $R_{N:P}$ of P:N:C:-O$_2$ = 1:16:122:172 (Takahashi et al., 1985). Phytoplankton production is limited by the availability of nutrients (phosphate, nitrate, ammonium or iron), light and temperature. For a detailed description of the standard dependencies of growth on light and temperature we refer to Ilyina et al. (2013).

For each nutrient, we determine a potential phytoplankton growth $G_{phy}$ with the corresponding half saturation constant for that nutrient separately and, then, use the minimum





$$
\begin{aligned}
G_{phy} &= \min(J(T, I(z))(F_{N_i})\,\mathrm{Phy}\,R_{N:P} \\
\text{with } F_{N_i} &= \frac{N_i}{(bk_i + N_i)} && \text{for i being P or Fe} \\
\text{and } F_{N_i} &= \frac{N_{\mathrm{NO_3}}\,\sigma_{inhib}}{bk_{\mathrm{NO_3}} + N_{\mathrm{NO_3}}} + \frac{N_{\mathrm{NH_4}}}{bk_{\mathrm{NH_4}} + N_{\mathrm{NH_4}}} && \text{for i being TIN} \\
\text{and } \sigma_{\mathrm{NH_4}} &= \left(\frac{N_{\mathrm{NH_4}}}{bk_{\mathrm{NH_4}} + N_{\mathrm{NH_4}}}\right) \Big/ \left(\frac{N_{\mathrm{NO_3}}\,\sigma_{inhib}}{bk_{\mathrm{NO_3}} + N_{\mathrm{NO_3}}} + \frac{N_{\mathrm{NH_4}}}{bk_{\mathrm{NH_4}} + N_{\mathrm{NH_4}}}\right) \\
\text{and } \sigma_{\mathrm{NO_3}} &= 1 - \sigma_{\mathrm{NH_4}}
\end{aligned}
$$

Phytoplankton favours the uptake of ammonium, if present, which is considered by the inhibition factor (Fennel et al., 2006)

$$
\sigma_{inhib} = \frac{1}{1 + N_{\mathrm{NH_4}}/bk_{\mathrm{NH_4}}}
$$

$J(T, I(z))$ is the maximal growth rate. It is depending on temperature (Eppley, 1972) and available photosynthetic radiation at the depth z (see Paulsen et al. (2017) for more details).

The process descriptions for grazing activities and bacterial decomposition are identical to Ilyina et al. (2013), but aerobic and anaerobic remineralization processes are updated. Aerobic remineralization of $F_{det}$ is a linear function of detritus concentration occurring only in water with an oxygen concentration above 2 mmol m$^{-3}$.

$$
\begin{aligned}
F_{det} &= \sigma_{o2}\lambda_{det}\mathrm{Det} && (5) \\
\text{with } \sigma_{o2} &= \frac{\mathrm{O_2}}{(\mathrm{O_2} + \mathrm{O_{2crit}})} && (6)
\end{aligned}
$$

$\sigma_{o2}$ slows down the remineralization rate at oxygen concentrations below O$_{2crit}$=20 mmol m$^{-3}$ (Table A1).

All nitrogen in organic material is remineralized to ammonium and, thereby, less oxygen is consumed than it is needed to remineralize organic matter to nitrate (Paulmier et al., 2009). In oxygenated water stepwise nitrification converts NH$_4^+$ and NO$_2^-$ back to NO$_3^-$. These processes are inhibited in the presence of light (Fennel et al., 2006).

$$
\begin{aligned}
O_{\mathrm{NH_4}} &= \lambda_{oxy\mathrm{NH_4}}\mathrm{NH_4^+}\,\sigma_I && (7) \\
O_{\mathrm{NO_2}} &= \lambda_{oxy\mathrm{NO_2}}\mathrm{NO_2^-}\,\sigma_I && (8) \\
\text{with } \sigma_I &= \frac{bk_I}{(bk_I + I(z))}
\end{aligned}
$$

In oxygen minimum zones (OMZs) , i.e.suboxic waters with oxygen concentrations below O$_{2crit}$, processes of dissimilatory nitrate reduction to nitrite ($DNRN$) or ammonium ($DNRA$), and heterotrophic denitrification ($NRN2$) degrade organic





material while consuming fixed nitrogen.

$$DNRN \quad = \quad (1 - \sigma_{ano2})\lambda_{dnrn}\frac{\mathrm{NO_3^-}}{bk_{dnrn} + \mathrm{NO_3^-}}\ \mathrm{Det} \tag{9}$$

$$DNRA \quad = \quad (1 - \sigma_{ano2})\lambda_{dnra}\frac{\mathrm{NO_3^-}}{bk_{dnra} + \mathrm{NO_3^-}}\ \mathrm{Det} \tag{10}$$

$$NRN2 \quad = \quad \lambda_{nrn2}\frac{\mathrm{NO_2^-}}{bk_{nrn2} + \mathrm{NO_2^-}}\ \mathrm{Det} \tag{11}$$

In the case of $DNRN$ and $DNRA$ intermediate steps of nitrate reduction to $\mathrm{NO_2^-}$ or $\mathrm{NH_4^+}$ might be oxidized back to $\mathrm{NO_3^-}$ in the presence of oxygen. Only $NRN2$ leads to a complete loss of fixed nitrogen while organic material is degraded.

The second pathway for complete loss of fixed nitrogen is the anaerobic oxidation of $\mathrm{NH_4^+}$ with $\mathrm{NO_2^-}$, anammox. Anammox bacteria combine ammonium and nitrite to dinitrogen and hereby produce small amounts of nitrate (Brunner et al., 2013).

$$Anam = (1 - \sigma_{ano2})\lambda_{anam}min(\mathrm{NH_4^+}, \mathrm{NO_2^-}) \tag{12}$$

The concentration of $\mathrm{NH_4^+}$ is generally low in the ocean which means that the magnitude of $Anam$ is mainly controlled by the availability of $\mathrm{NH_4^+}$. $\sigma_{ano2}$ is defined to be equal 1 at an oxygen concentration larger than the threshhold value $\mathrm{O_{2crit}}$ and it decreases with decreasing oxygen according to

$$\sigma_{ano2} = \frac{\mathrm{O_2}}{\mathrm{O_{2crit}}} \tag{13}$$

This allows that $DNRN, DNRA$, and $Anam$ start to occur in suboxic waters as it has been observed (Kalvelage et al., 2013). Complete denitrification $NRN2$ is restricted to oxygen concentrations below 2 mmol m$^{-3}$. All rates $\lambda$ and half saturation constants $bk$ are given in Table A1.

Please note, that all processes described above are active in the water column and the sediment. Diffusive fluxes for all nitrogen components at the water-sediment boundary are also included.

### 2.2.2 Air-sea gasexchange of ammonia

Ammonia (NH$_3$) is a soluble gas and its air-sea transfer is generally considered to be under gas-phase control (Liss, 1983). The concentration of NH$_3$ is determined by the concentration of $\mathrm{NH_4^+}$, the dissociation coefficient for ammonium, which is a function of temperature ($T$), salinitiy ($S$), and the ionic strength of sea water H$^+$.

$$\mathrm{NH_3} = \mathrm{NH_4^+}\frac{K_a}{K_a + \mathrm{H^+}} \tag{14}$$

with

$$K_a = 10^{-pK_a}$$

$$pK_a = -0.467 + (0.00113S) + (2887.9/T)$$

The constants in Eq.(14) are taken from Johnson et al. (2008). The flux of ammonia across the ocean-atmosphere interface is a function of the gas-phase transfer velocity $k_g$ and the concentration gradient for NH$_3$ between ocean and atmosphere. Here,



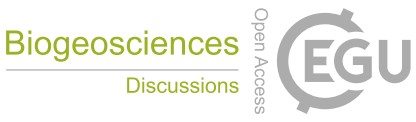

we assume that the atmospheric concentration of $NH_3$ is neglectable (atmospheric concentrations are generally less than 1 nmol m$^{-3}$, given no terrestrial sources) and the flux is only a function of the sea water concentration times the dimensionless coefficient of Henry's law for ammonium $K_H$.

$$\Psi_{airsea} = -k_g K_H NH_3 \tag{15}$$

$$k_g = u/(770 + 45 \, RMM^{1/3})$$

$\quad K_H^{-1} = 17.93 \, (T/273.15) \, e^{4092/T - 9.70}$

RMM denotes the molecular mass of ammonia = 17.03 g mol$^{-1}$. $u$ is the wind speed (m s$^{-1}$). Again, the fit for the dissociation constant is taken from Johnson et al. (2008).

### 2.2.3 Nitrogen fixation

The loss of fixed nitrogen by processes of denitrification and anammox in the OMZ is compensated by the fixation of elementary nitrogen due to diazotrophs living in the sunlite zone surface ocean. Current estimates of global nitrogen fixation (163 Tg N yr$^{-1}$) in combination with atmospheric deposition and river input (37 Tg N yr$^{-1}$) are of the order of estimates for loss of fixed nitrogen in the water column and the sediment (200 Tg N yr$^{-1}$) (Wang et al., 2019). In HAMOCC, a prognostic treatment of the growth of diazotrophs was introduced by Paulsen et al. (2017) which results in a more realistic spatial distribution N$_2$ fixation.

Global fixation (135.6 Tg N yr$^{-1}$) approximately balances global denitrification in the water column and the sediment (141.9 Tg N yr$^{-1}$) for present day conditions. Here, we are briefly summarizing the major features of this prognostic treatment and we refer to the full description of this model development to Paulsen et al. (2017).

As for bulk phytoplankton, diazopthrophs live in the upper ocean controlled by the avalability of nutrients, light and temperature. In contrast to bulk phytoplankton, diazothrophs, especially the here considered *Trichodesmium*, have a distinct optimal

temperature range in which they can grow. This limits their distribution to the zonal band of 40°N to 40°S (Paulsen et al., 2017). Diazotrophs are described to have a smaller maximal growth rate than bulk phytoplankton, but they have the same biochemical composition of their organic tissue following the fixed Redfield ratio. In regions with sufficient nutrient supply diazotrophs have to compete with bulk phytoplankton. Their ability to fix N$_2$ for their N supply allows them to also grow in regions which are already depleted with respect to bioavailable nitrogen compounds. The mortality of diazotrophs is considered a first-order

process with a constant decay rate. The major part of the organic material is directed to the detritus pool (90 %) and subject to export by gravitational sinking. The remaining part is consider as dissolved organic matter which decays with a constant rate (Paulsen et al., 2017; Ilyina et al., 2013).

## 2.3 Model setup and transient simulations

Our goal is to run a simulation with a consistent forcing over the 20th century allowing us to analyse decadal trends. To this end, we need to perform a spin-up run which, in its final state, has a low artificial model drift and shows a reduced initial shock



when applying the transient ERA20C forcing.

To achieve starting points for the transient simulation for both model setups, a spin-up procedure was done in several steps. First, we forced both models with the same atmospheric boundary conditions from a climatology based on the second ECMWF
Re-Analysis project (ERA-40) (Simmons and Gibson, 2000) from which a mean annual cycle in daily resolution for relevant parameters like wind stress, heat and freshwater fluxes was calculated by Röske (2006)(OMIP forcing). Salt relaxation to PHC3.0 data (Steele et al., 2001) with a time constant of 1 month is used globally at the surface. The coarse resolution of higBUS in the region of deep water formation in the North Atlantic leads to a weak meridional overturning and made it necessary to also apply a 3-d field salt nudging to PHC3.0 north of 60° N with a time constant of 1 month.

For GR15, being the "work horse" of the Max Planck Institute for several decades, we used existing restart files for MPIOM and HAMOCC which have been spun up for several millennia. The initial condition for the physical field of higBUS was started from rest based on temperature and salinity fields from the World Ocean Atlas (Boyer et al., 2018). We run only the physical part of the model for approx. 200 years before it was coupled to HAMOCC. The higBUS restart file for the biogeochemistry was taken from the GR15 configuration and interpolated onto the new grid. Coarse resolution models, in general, overestimate
the oxygen minimum zone (OMZ) in the equatorial Pacific (e.g. Cocco et al. (2013)). In our global set-up, we avoid an impact of this unrealistic large OMZ in the Pacific on the resulting tracer distributions in the Benguela region by nudging the simulated oxygen concentration to observations (WOA,Boyer et al. (2018)) in the equatorial Pacific. This is only done in case simulated oxygen concentrations drop below a threshold value (20 mmol m$^{-3}$). The nudging coefficient increases linearly with decreasing oxygen concentration and would correspond to a time constant of 4 months at a theoretical value of 0 mmol
m$^{-3}$. For consistency, the nudging of oxygen is applied in both model versions, higBUS and GR15. Both model set-ups were integrated with the new features of the nitrogen cycle for 1000 years with this climatological forcing.

In a second step, we switched to the ERA20C forcing, which is ECMWF's first atmospheric reanalysis of the 20th century, from 1901-2010 (Poli et al., 2016). We converted the data from a 3-hourly to a 6-hourly forcing. The continental runoff was adopted from the climatological OMIP forcing (Röske, 2006). For the spin-up phase we use only the years 1910 to 1939 of the
ERA20C forcing as a cyclic forcing. We prescribe a fixed atmospheric pCO$_2$ corresponding to the year 1900 (295.7 ppmV). After about 960 years we started the transient simulation from 1901 to 2009 with the corresponding ERA20C forcing and increasing atmospheric pCO$_2$ following the historical reconstruction (https://data.giss.nasa.gov/modelforce/ ghgases).

We are aware of the fact that a good spatial resolution of the local wind stress field is essential to capture some of the main characteristics of the Benguela upwelling system (e.g. Desbiolles et al. (2014), Junker et al. (2015)). The atmospheric model
(T159) used for the ECMWF Re-analysis project ERA20C has a horizontal resolution of approx. 125 km. The comparison of the wind stress curl product from ERA20C with a higher horizontal resolution product (ERA-Interim 075 based on T255, Dee et al. (2011)) show a very similar pattern with stronger maxima and a narrow zonal band of negative wind stress curl along the African coast (not shown). Even though the ERA-Interim 075 product might give a better representation of the local wind forcing, it covers only approx. 30 years (1979 - present) and therefore includes already a strong anthropogenic signal. Other re-
analysis products, such as NOAA-CIRES-DOE 20th Century Reanalysis version 2 (http://www.esrl.noaa.gov/psd/data/20thC_ Rean) are ruled out due to their even coarser spatial resolution (2x2 deg).





## 2.4 Additional Diagnostics

To gain more insight into the flow pattern and water mass composition we introduce artificial dye tracers. Dyes are passive tracers and follow the identical hydrodynamical processes as biogeochemical tracers. They have no sinks or sources within the
ocean interior. Their boundary condition is set to 1 at the surface within their source region and set to 0 at the surface outside of this region.

$$\frac{\partial D_i}{\partial dt} + \nabla \boldsymbol{U} D_i - D D_i \;\;=\;\; 0 \tag{16}$$

$$\text{with} \quad \frac{\partial D_i}{\partial dt} \;\;=\;\; 1 \quad \text{at the surface within source region of } D_i$$

We define 9 source regions (Fig. 3) to cover the whole model domain (without any overlaps). The relative contribution of a dye concentration to the sum of all dye concentrations within a grid box tells us the regional origin of the water mass. In addition, we include a linear age tracer. It is set to zero at the ocean surface and treated as the dye tracers while constantly ageing in the ocean interior.

Furthermore, we track the advection of oxygen, phosphate, and detritus to investigate the eddy contribution versus mean
flow transport. In general, the eddy transport for a tracer B at each geographical position can be estimated from the relationship

$$\overline{\boldsymbol{U}B} = \overline{\boldsymbol{U}}\,\overline{B} + \overline{\boldsymbol{U}'B'} \tag{17}$$

$\boldsymbol{U}$ corresponds to the flow directions in zonal, meridional and vertical direction (u,v,w respectively). By substracting the temporal average of the product of the monthly mean velocity $\overline{\boldsymbol{U}}$ times the temporal average of the monthly mean tracer concentration $\overline{B}$ from the temporal average of the monthly mean product $\overline{\boldsymbol{U}B}$ we obtain the nonlinear part of the flow field
which can be interpreted as eddy-induced transport (von Storch et al., 2016), but which includes interannual variability as well. We use the term "eddy-induced" for $\overline{\boldsymbol{U}'B'}$ even in the coarse resolution GR15, where meso-scale structures are not resolved. Our analysis of meso-scale structures is based on monthly mean values because of the strong seasonality in the BUS.

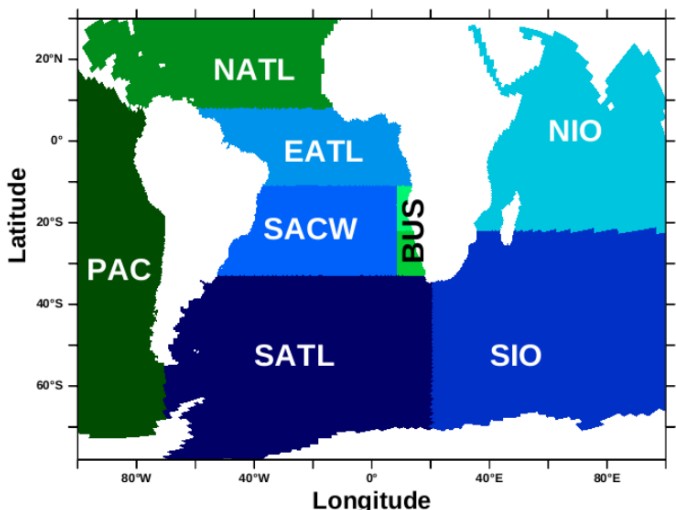

**Figure 3.** Schematic view on the different source regions for nine dye tracers. NATL=North Atlantic (>10° N), EATL=Equatorial Atlantic (10° S - 10° N), SACW=South Atlantic Central Water (10° S - 33° S, coast of South America to 8.5° E), SATL=South Atlantic (33° S - 90° S, 70° W - 20° E), SIO=Southern Indian Ocean (23° S - 90° S, 20° E - 120° E), NIO=Northern Indian Ocean (<23° S), PAC=Pacific, BUS=Benguela upwelling system with two regions from 8.5° E to the coast, separated at 22° S

## 3 Model results compared to observations

To evaluate the simulated circulation and the biogeochemistry in the BUS we show results of the averaged last 50 years of the simulation with the ERA20C forcing (1960-2009). For individual observational products, we might choose a different time period for averaging which will be indicated. Results of GR15 are shown to illustrate the added value of the high resolution set up.

### 3.1 General circulation pattern

The large scale circulation pattern and long term mean transports through major passages are similar between higBUS and
GR15 and compare reasonably well to observations (Table 1). Due to the relative coarse resolution of the northern part of the Atlantic Ocean and the Arctic the Atlantic Meridional Overturning Circulation (AMOC) in higBUS is rather low.

        For the water mass composition of the BUS two prominent circulation patterns are relevant. The southern part of the BUS is strongly influence by the Agulhas current and its retroflection. Warm and salty water is spreading around the southern tip
of Africa, turns southward and flows back into the Indian Ocean as indicated by the barotropic streamfunction of higBUS (Fig. 4a). Part of this water, transported from the Indian Ocean northward into the Atlantic Ocean, feeds the Bengula current system (Agulhas leakage, Biastoch et al. (2009)). It has been already shown that eddy permitting model resolutions give a better representation of the Agulhas retroflection (e.g. Jungclaus et al. (2013)). The barotropic streamfunction shows two distinct cells





**Table 1.** Long-term mean transport through major passages (Sv) from last 50 yr of the transient simulation for higBUS and GR15 and from observational estimates

| Section | higBUS | GR15 | Observations | Ref |
|---|---|---|---|---|
| Drake Passage | 199±3.7 | 196±3.5 | 137±8 | Cunningham et al. (2003) |
| Indonesian Throughflow | 9.6 ±1.5 | 11.3±1.4 | 11.6-15.7 | Gordon et al. (2010) |
| Bering Strait | 0.33±0.1 | 0.49±0.08 | 0.8-1 | Woodgate et al. (2006) |
| Mozambique Channel | 14.2±2.2 | 14.8±2.9 | 5-26 | DiMarco et al. (2002) |
| AMOC at 26.5°N | 12.4±0.99 | 15.0±1.2 | 18.7±2.1 | Kanzow et al. (2010) |

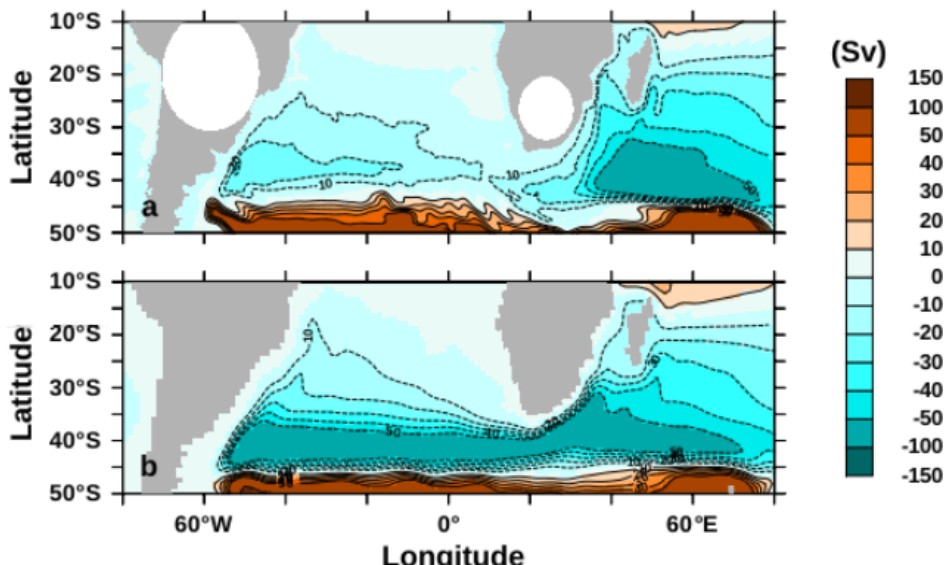

**Figure 4.** Climatological mean barotropic streamfunction (Sv) of higBUS (a) and GR15 (b). Note the non-linear color scale. Contour lines are at -50 to 50 with an increment of 10.

in the zonal band of 30-45° S in higBUS being in line with observation based estimates (Wunsch, 2011). In contrast, GR15
shows only one "super cell". Models with coarser resolution tend to have a more laminar flow field with a higher transport of Indian Ocean water into the Atlantic and a stronger Agulhas leakage at the tip of Africa (e.g. Marcello et al. (2018), Fig.1c ). This is underlined by the distribution of the dye tracer from SIO at 50-150 m in the South Atlantic (Fig. 5).

In higBUS, SIO water reaches the South Atlantic and is carried northward with the Benguela current without much westward spreading. Water from the SIO is the dominant fraction of the water mass composition south of 30° S at 50-150 m in our
simulation. Here, at the southern tip of Africa, water from the Agulhas leakage and South Atlantic water mix to form Eastern South Atlantic Central Water (ESACW, Mohrholz et al. (2008)) as concluded from observations. The simulated dye tracer





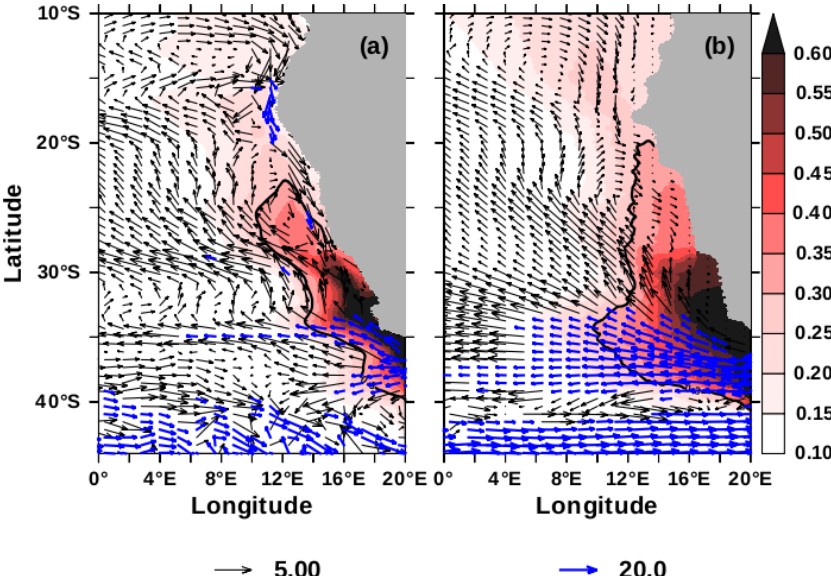

**Figure 5.** Climatological mean contribution of Southern Indian Ocean water at a depth of 50 to 150 m indicated by the fraction of the dye tracer originating in the SIO region (colour) for higBUS (a) and GR15 (b). Contour line of 0.3 is highlighted (black). Overlaid are velocity arrows averaged over 50-150 m. Black arrows are scaled to 5 cm/s, blue arrows are scaled to 20 cm/s; see reference arrows below the figure.

distribution along a zonal section at 30° S shows a small contribution from SATL and a dominant fraction of SIO water (Fig. A1a,b), which is in line with the current concept of the origin of ESACW. Interestingly, in higBUS there is also a small contribution of SACW water on the shelf.

This water from SACW enters the BUS region at its northern boundary (Mohrholz et al., 2008) and is advected by a strong poleward coastal current, very prominent at 23° S (Fig. 6c), which pushes the branch of the Benguela current westward and prevents the spreading of SIO water along the coastal shelf (Fig. 5a, and Fig. A2). In higBUS, the poleward current is more intense and narrower than in GR15 with climatological mean velocities of more than 6 cm/s in the core at 100-200 m depth. Close to the coast, we find a northward current which was also identified from current measurements at a mooring station at

23° S and 14° E (see Mohrholz et al. (2008), Fig. 14). Fig. 6a,b already give some indications of a more realistic cross shelf exchange in higBUS where we find a zonal mean velocitiy of > 2 cm s$^{-1}$ at the shelf break.

    A water mass analysis by Mohrholz et al. (2008) for the same location (mooring data in 120 m at 23° S, 14° E) also supported a dominant fraction of SACW compared to ESACW (i.e. equivalent to the sum of SIO and SATL in our setup, see Fig. 3). The study by Tim et al. (2018) tracking Lagrangian particles found a similar distribution with a high contribution of SIO water

in the southern BUS (between 27° S to 34° S) and a lower presence of SIO in the northern part (between 17° S and 27° S). Results of higBUS are consistent with these estimates, while GR15 is not able to resolve these complex circulation structures showing a different pattern with higher SIO and SATL fractions in the northern BUS (Fig. 5,Fig. A2).


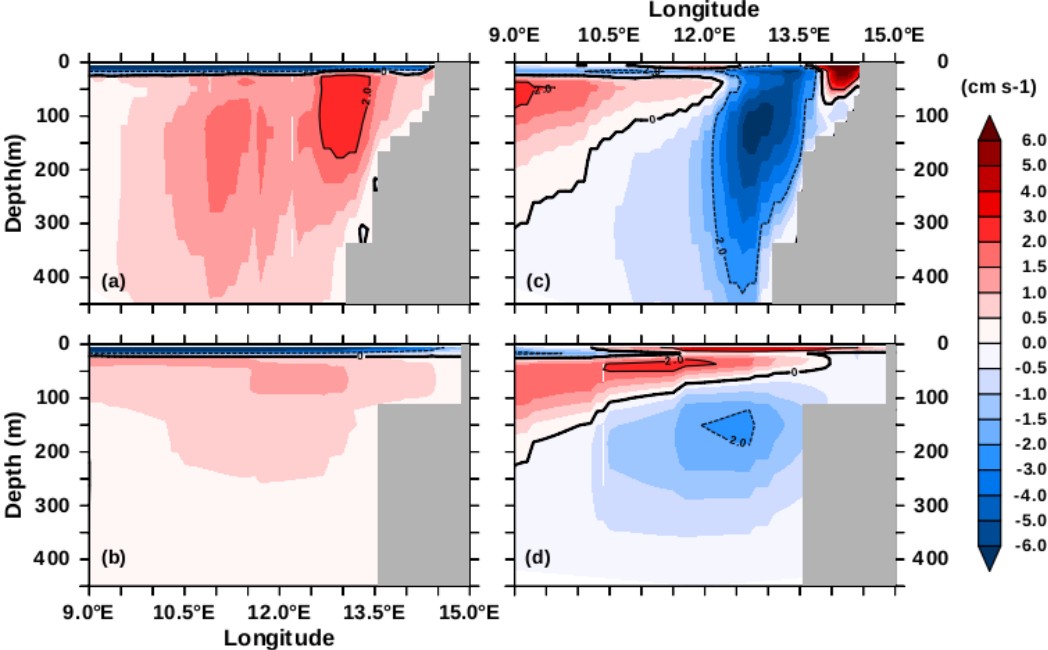

**Figure 6.** Climatological mean velocities (in cm s$^{-1}$) in zonal (a,b) and meridional (c,d) directions at 23° S for higBUS (a,c) and GR15 (b,d) of the upper 450 m. Negative numbers indicate a westward/southward velocity. Contour lines at -2., 0 , 2.. Note the nonlinear color scale.

Besides differences in the water mass composition higBUS shows, in general, much higher surface velocities with a pattern similar to observation-based estimates (Ocean Surface Current Analyses Real-time (OSCAR), ESR (2009)) (Fig. 7). Especially,

south of 32° S the velocities components reflect the pattern already discussed for the barotropic streamfunction. The meridional velocity component shows a pronounced dipole structure southwest of the African tip which is also seen in OSCAR. Part of the Agulhas current is branched off close to the coast (see a strong northward component in Fig. 7d,e) influencing primarily water masses of the southern BUS. In the northern BUS (north of 28° S) we see a pronounced poleward surface current in higBUS which is partly also identifiable in OSCAR (Fig. 7 d,e). In contrast, GR15 misses most of these features. A comparison of

the sea surface height anomaly also underlines the very different dynamical behaviour of the eddy-permitting higBUS and the coarse resolution GR15 (see discussion in the Appendix and Fig. A3).

The heterogeneous spatial pattern of the surface velocity components in higBUS results in a heterogeneous upwelling pattern at 100 m (Fig. 8) with distinct and intense upwelling centers. Major upwelling centers are located at 16-18° S (Cunene) and 26-28° S (Lüderitz) as found in other studies (Veitch et al., 2010). In GR15 most of the upwelling cells in the near shore band

are missing. The effect of the wind-induced offshore Ekman transport, which causes the upwelling, leads to a local decrease in the near surface temperature (Fig. 9). Accordingly, in higBUS, coastal temperatures south of 22° S are in better agreement with the near surface temperature reanalysis from ORAS5 (Zuo et al., 2019), although still with a tendency of too warm temperatures. Especially between 26° S and 22° S, higBUS is warmer with a reduced zonal gradient compared to ORAS5.


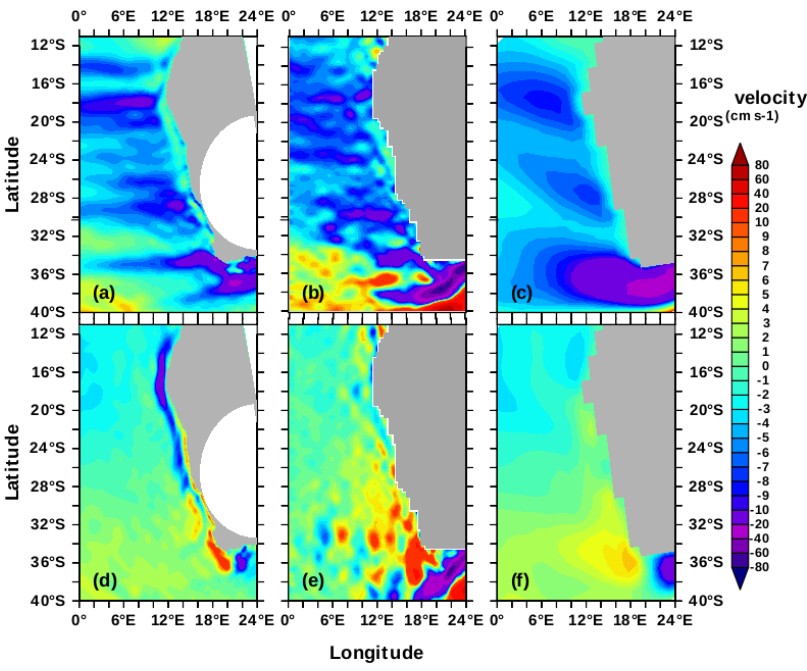

**Figure 7.** Mean surface velocities (cm s$^{-1}$) of the zonal (a,b,c) and meridional (d,e,f) component from higBUS (a,d), an estimate derived from satellite observations (OSCAR, b,e), and GR15 (c,f) for 2000-2009. Note the no-linear color scale.

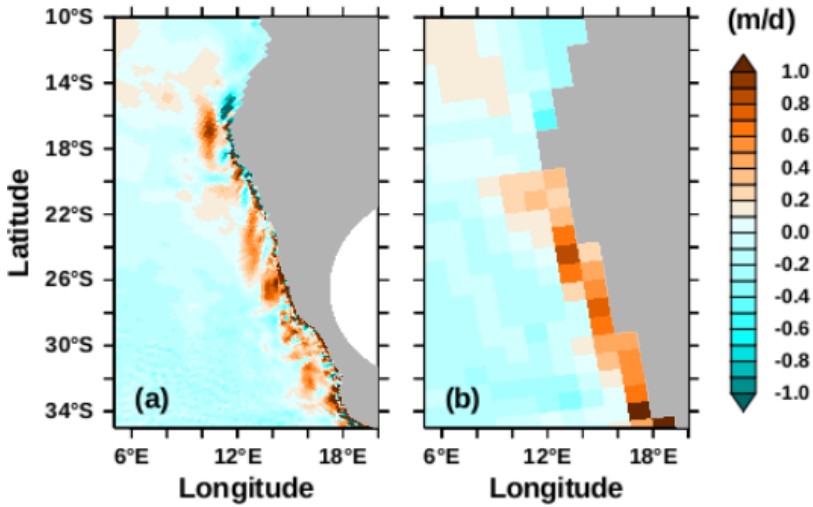

**Figure 8.** Climatological mean upwelling velocity at 100 m averaged over September to December for higBUS (a) and GR15(b). Units m d$^{-1}$





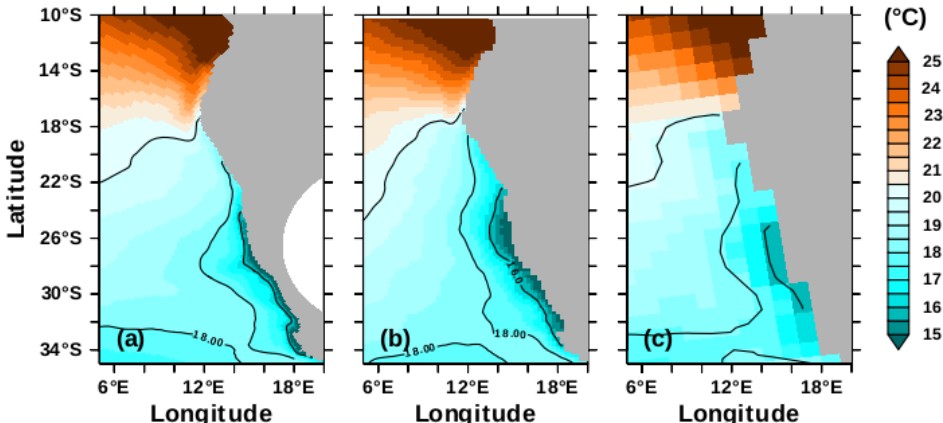

**Figure 9.** Mean ocean temperature in the upper 10 m from higBUS (a), reanalysis data (b, ORAS5, Zuo et al. (2019)), and GR15 (c) for 1980-2009 given in °C. Contour lines are given for 16°C and 18°C.

These biases demonstrate the importance of the horizontal resolution of the atmospheric wind stress forcing on the simulated
local near surface temperatures in the BUS (Milinski et al., 2016) which could also explain a reduced Ekman pumping with underestimated supply of cold water. North of 18° S a narrow warm surface current coming for the north is captured in higBUS as found in the reanalysis. Again, GR15 does not produce the cold temperature pattern along the African coast.

To summarize, the higBUS simulation shows that the incorporation of meso-scale activity enables a reasonable representation of the major hydrodynamical pattern of the BUS. Discrepancies as described above still exist due to the relative coarse
resolution of the atmospheric forcing, a potentially still too low resolution of shelf areas and the neglection of tidal forcing. However, as we will see in the following sections, the more realistic circulation regime in higBUS has a significant impact on the simulated biogeochemical distributions and the transient response to anthropogenic-induced changes.





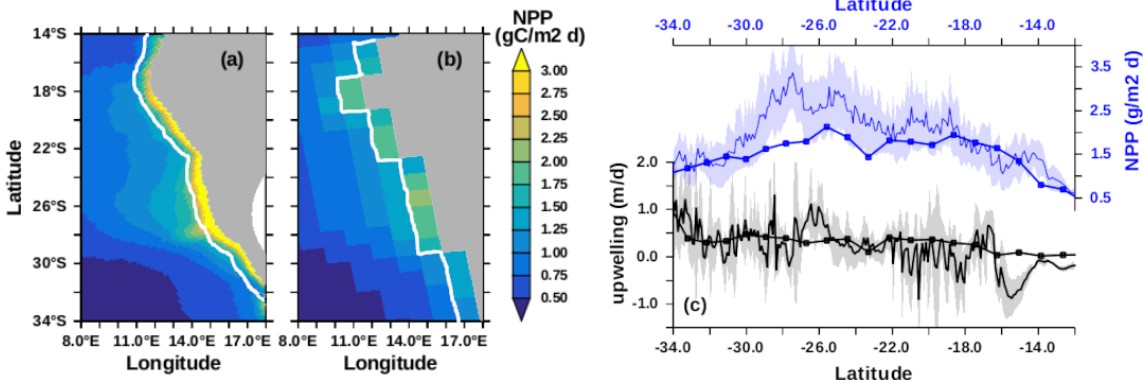

**Figure 10.** Climatological mean vertical integrated net primary production (gC m$^{-2}$ d$^{-1}$) for higBUS (a) and GR15 (b). Panel (c): zonal averages of a band along the African west coast (indicated by whites line in a,b) of the climatological mean vertical integrated net primary production (blue, gC m$^{-2}$ d$^{-1}$) and the climatological mean upwelling velocity (black, m d$^{-1}$) for higBUS (thin lines) and GR15(thick lines with symbols). The shaded area indicates the spatial variability within this band. For GR15 the value corresponds to the first grid box west of the coast while for higBUS it is a zonal average over 1.5° west of the coast line (approx. indicated by white lines in (a) and (b).

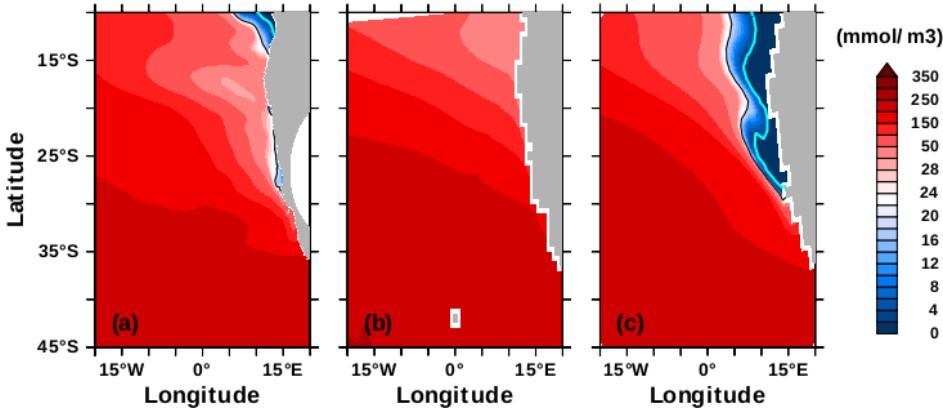

**Figure 11.** Climatological mean O$_2$ concentrations between 200-400 m (mmol m$^{-3}$) of higBUS (a), from observational data (b, Schmidtko et al. (2017), and of GR15 (c). Contour lines indicate O$_2$ concentrations of 20 mmol m$^{-3}$ (black) and 2 mmol m$^{-3}$ (lightblue) .

## 3.2 Biogeochemical pattern

The pattern of biogeochemical variables in the BUS are primarily controlled by water mass composition and upwelling struc-
tures (Hutchings et al., 1995; Williams and Follows, 2003). Thus, the pattern of net primary production (NPP) resembles the spatial distribution of upwelling with high NPP rates along the coast in higBUS (Fig. 10).

In the upwelling center of Lüderitz (26-28° S), the NPP in higBUS is significantly higher (times 1.5-2) in the near shore band (about 155 km wide, Fig. 10c) than in GR15. Off Namibia at 23° S we also find a stronger NPP while the climatological mean upwelling rate is comparable between higBUS and GR15 at this location.





High NPP can only be achieved by high nutrient supply to the production zone. Subsequent high export rates of organic material potentially lead to high remineralization induced oxygen consumption at depth. This is reflected in the low oxygen concentrations on the shelf in higBUS (Fig. 11). The offshore regions show very similar concentrations as found in the climatology from Schmidtko et al. (2017). Note, that the shelf areas of the BUS might not be well represented in the climatology from Schmidtko et al. (2017). In GR15, we find a broad band of low $O_2$ concentrations along the entire coast of the northern

BUS (north of 30° S), stretching out into the open ocean. Moreover, the $O_2$ concentrations in this 2-10° broad band are lower than the threshold value for nitrate reducing processes (Fig. 11c, black line) and even pass the critical $O_2$ concentration of 2 mmol m$^{-3}$ for complete denitrification (Fig. 11c, lightblue line). This is a first indication that different biogeochemical processes are at play in GR15 and higBUS.

    To evaluate the performance of higBUS and to illustrate the very different near shore conditions between higBUS and GR15,

we focus on the well studied area off Namibia (box limits at 22° - 24° S and 12° E - coast, see e.g. Emeis et al. (2018)). We compare model data to mean concentrations calculated from data of 4 GENUS cruises (Lahajnar (2012a, b, 2015a, b)) for the same region. Cruise data and model data were sampled and averaged to a 0.5° by 0.5° grid with vertical resolution of the model. Monthly mean model data are taken only where observational data are available.

    Despite higher local NPP (+12 %) and, thus, higher export of organic material, we find higher oxygen concentrations at depth

in higBUS than in GR15 (Fig. 12a). $O_2$ concentrations in higBUS are lower than the observations from 4 GENUS campaigns (Lahajnar, 2012a, b, 2015a, b), but within the spatial variability. Most important, the mean oxygen concentration is above 20 mmol m$^{-3}$, which is our defined threshold value for the occurrence of nitrate reduction processes. The nitrate and nitrite profiles of higBUS fit very well to the observation, indicating a prevailing remineralization of organic material on dissolved oxygen or a fast reoxidation of products from nitrate reduction processes. Simulated rates of the first step for nitrate reduction

($DNRN$) and nitrification of $NH_4^+$ and $NO_2^-$ fit well to observed rates from other coastal OMZs (Kalvelage et al., 2013) and model studies (Azhar et al., 2014). Nitrogen loss processes as denitrification ($NRN2$) or anammox ($Anam$) mainly occur between 100-150 m depth at very low rates in higBUS. In contrast, in GR15 we see high rates of the nitrate loss processes ($DNRN$, $NRN2$, and $Anam$) occurring in a very low oxygen environment (Fig. 12a) and resulting in $NO_3^-$ concentration of only 15 % of the observed concentration at 400 m (Fig. 12c). Additionally, $NO_2^-$ accumulates to more than the 40-fold observed

value. The lack of oxygen prevents nitrification back to $NO_3^-$.

    GR15 also shows the typical feature of artificial nutrient trapping, that is present in upwelling regions in many coarse resolution models indicated by accumulated phosphate concentration (Najjar et al., 1992; Aumont et al., 1999; Dietze and Loeptien, 2013). The local N:P ratio below 50 m is significantly lower in GR15 than observed or simulated in higBUS (Fig. 14). Below 200 m a reversed linear N:P relationship clearly indicates that strong N-loss processes are in full swing. Given

the prescribed fixed N:P ratio for organic matter production in HAMOCC, it is obvious that the upwelling water in GR15 can supply only a limited NPP due to the lack of nitrate (Fig. 10).

    Nitrogen fixation, the marine process that, in general, compensates the N-loss from denitrification and anammox in the ocean, is not active in the BUS region in both model resolutions. Temperature limitations restrict the growth rate of diazotrophs, which primarily resembles the growth characteristics of *Trichodesmium* (Paulsen et al., 2017), in HAMOCC. However, for the entire





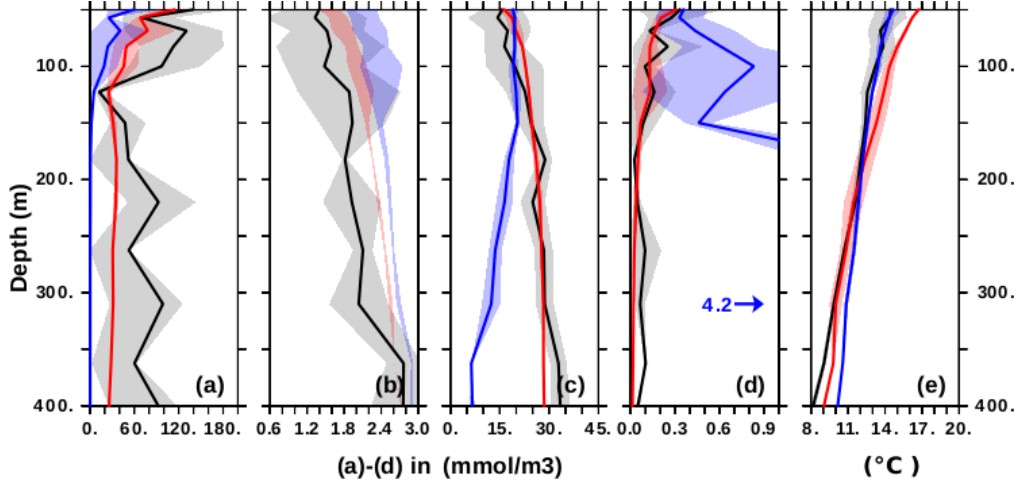

**Figure 12.** Mean profiles of oxygen (a), phosphate (b), nitrate (c), nitrite (d) (all in mmol m$^{-3}$) and temperature (°C) at 22-24° S and 12° E-coast of October to February of the years 2000-2009 for higBUS(red) and GR15(blue). Overlaid are mean concentrations (black) calculated from data of 4 GENUS cruises (Lahajnar (2012a, b, 2015a, b) ) for the same region. Depth range is 50-400 m. Shaded area indicate the spatial standard deviation of observations or model data. Arrow in (d) indicates the depth and the magnitude of the maximum NO$_2^-$ concentration in GR15.

BUS region, there is no published observational evidence for the presence of *Trichodesmium* (Emeis et al., 2018) in line with our simulations. Furthermore, no or very low nitrogen fixation was detected during different measuring campaigns (Staal et al., 2007; Wasmund et al., 2015).

     This comparison to observations gives us the confidence that the N-cycle is well represented in higBUS, based on sufficient oxygen supply to keep ocean water in the BUS region well above O$_2$ levels at which nitrogen loss processes start. In contrast,
hydrodynamical processes in GR15, especially the water mass ventilation, are too weak to stabilize O$_2$ levels above our nitrate reduction threshold (O$_{2\ crit}$ = 20 mmol m$^{-3}$). The consequences are too low concentrations of oxygen and nitrate, higher than observed phosphate levels, and unrealistic high concentrations of nitrite.

### 3.3   Driving forces of O$_2$ supply

     The physically induced contribution for different O$_2$ pattern of higBUS and GR15 is primarily lateral transport, especially in
the zonal, onshore direction (Fig. 15). The O$_2$ transport in several zonal bands is towards the coast in higBUS. Around 17° S the zonal transport is away from the coast, but here, we find the largest meridional O$_2$ transport coming from the North induced by the poleward coastal undercurrent. All of these features are missing in GR15 due to the different horizontal circulation pattern (Fig. 5) and the depleted O$_2$ concentrations in a broad off shore region (Fig. 11). In addition, climatological mean annual maximum mixed layer depths (Fig. A4) in higBUS are deeper in the near shore coastal band. Therefore, areas on the
shelf and in its vicinity are better ventilated than in GR15.


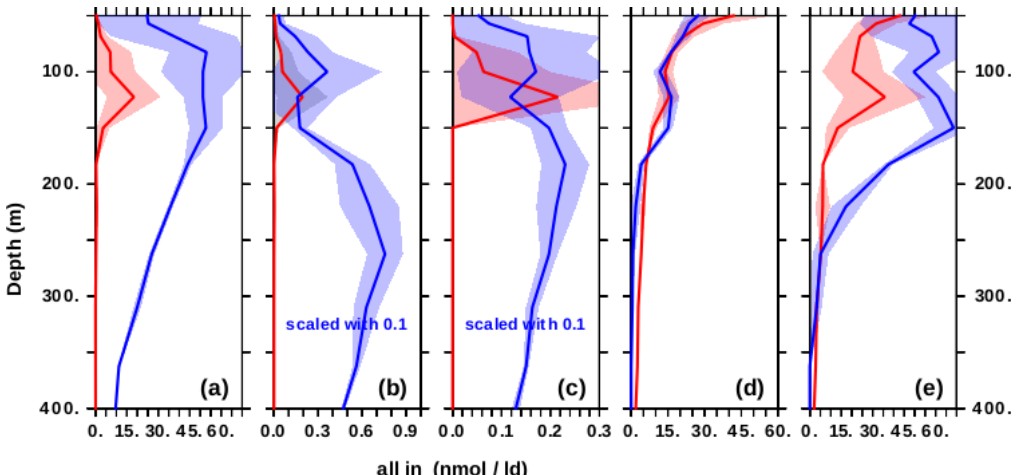

**Figure 13.** As Fig. 12, but for mean profiles of rates of dissolved nitrate reduction to nitrite ($DNRN$) (a), anammox ($Anam$) (b), nitrite reduction to $N_2$ ($NRN2$) (c), niftrification of $NH_4^+$ ($O_{NH_4}$) (d), and nitrification of $NO_2^-$ ($O_{NO_2}$) (e) (all in nmol $l^{-1}$ $d^{-1}$) for higBUS (red) and GR15 (blue). In (b) and (c) results for GR15 are multiplied with 0.1.

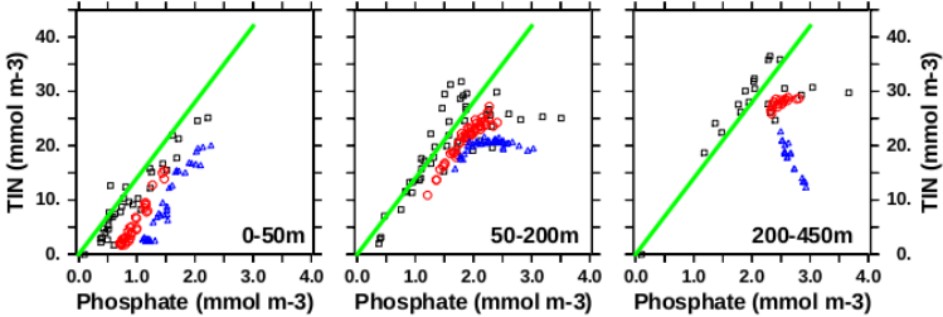

**Figure 14.** Phosphate versus total inorganic nitrogen (TIN, sum of nitrate, nitrite and ammonium) based on data shown in Fig. 12 from 4 Genus cruises Lahajnar (2012a, b, 2015a, b)(black symbols), higBUS (red), and GR15 (blue) for the depth ranges 0-50 m, 50-200 m and 200-450 m. Green line depicts the N:P ratio of 16:1 which is used for organic matter production in HAMOCC.

A closer look at the zonal section along 23° S shows that the climatological mean transport of $O_2$ is clearly eastward, i.e. towards the coast in the upper 400 m in higBUS (Fig. 16a). Only in the upper most 20-30 m the wind induced circulation transports oxygen away from the coast. The eddy-induced $O_2$ transport east of 11.5° E is also onto the shelf with a maximum between 50-150 m on the order of additional 10 % of the mean $O_2$ transport (Fig. 16b). Thus, the meso-scale structures at this

latitude contribute to the ventilation of the shelf. In GR15, we find a similar pattern, but the calculated climatological mean $O_2$ transport is much lower (Fig. 16c). This is due to overall smaller zonal velocities in GR15 (Fig. 6b), but also due to the more extended OMZ at 200-400 m (Fig. 11c) with lower absolute $O_2$ concentrations. Interestingly, the sign of the "eddy-induced"





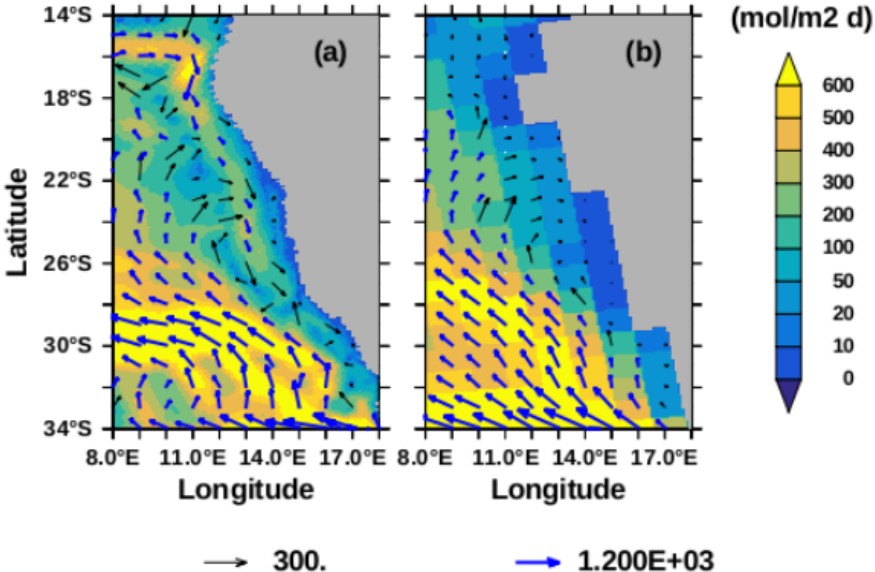

**Figure 15.** Mean lateral transport of oxygen ($(\overline{u\mathrm{O}_2}^2 + \overline{v\mathrm{O}_2}^2)^{0.5}$ in mol m$^{-2}$ d$^{-1}$) for 1960-1979 averaged over the depth range 50-150 m for higBUS (a) and GR15 (b). Note, the nonlinear color scale. Overlaid are transports directions for the same depth range; black arrows are scaled to 300 mol m$^{-2}$ d$^{-1}$, blue arrows are scaled to 1200 mol m$^{-2}$ d$^{-1}$; see reference arrow below the figure.

transport is negative (i.e. a westward O$_2$ transport). Thus, the temporal variability expressed by $\overline{u' \mathrm{O}_2'}$ rather lowers the zonal mean O$_2$ transport in GR15.

As already mentioned, the NPP on the shelf of higBUS is 3-4 fold higher than in GR15 (Fig. 17a). In addition, the zonal transport of detritus follows the zonal circulation with a strong eastward component (Fig. 17b). Thus, there is a high turnover of organic material which is reflected in the O$_2$ consumption (Fig. 17e). Highest rates are found in the upper 50 m and indicate primarily the O$_2$ demand by grazers, but aerobic remineralization occurs throughout the upper 300 m of the water column. As seen for the station data (Fig. 12a), climatological O$_2$ concentrations stay well above O$_{2crit}$. Due to the high NPP, the air-sea

flux of O$_2$ shows an outgassing almost over this entire section (Fig. 17d). An exception is only a small area close to the coast, which show an invasion of atmospheric O$_2$. Upwelling of nutrient rich water produces a high NPP, but this water is also cold and has, thus, a high solubility. Overall, the air-sea exchange rather plays a minor role in O$_2$ supply of the shelf at 23° S in higBUS. Again, the spatial pattern in GR15 is quite different with a lower NPP (Fig. 17a) and lower detritus transport onto the shelf (Fig. 17c). Still, O$_2$ concentrations are depleted below O$_{2crit}$ (Fig. 17f) on the shelf. The upper boundary of an OMZ is

found in 50 m depth on the shelf and at 200 m towards the west, both in contradiction with observations. The entire section east of 12.5° E is an atmospheric O$_2$ sink as a result of upwelling of O$_2$-depleted water and the lack of biological induced O$_2$ production (Fig. 17d).

To summarize, higBUS achieves a much better agreement with observations for the biogeochemical realm. This is true especially for the nitrogen components, NO$_3^-$, NO$_2^-$, and NH$_4^+$. Their concentrations are highly depending on the O$_2$ concentration.



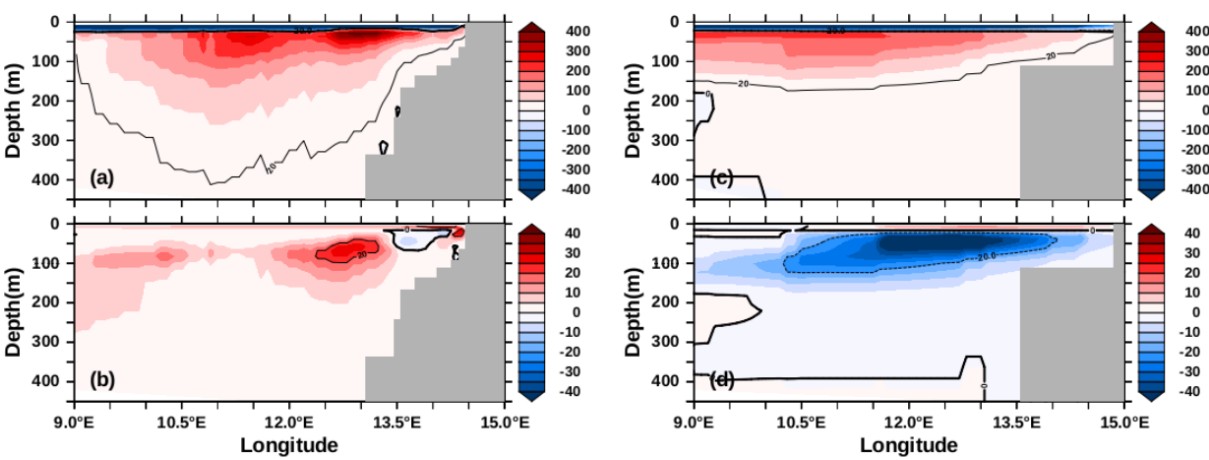

**Figure 16.** Calculated climatological mean zonal transport of oxygen ($\overline{u}\,\overline{O_2}$ in mol m$^{-2}$ d$^{-1}$) at 23° S for higBUS (a) and GR15 (c) of the upper 450 m. Positive numbers indicate eastward transport. Eddy-induced zonal mean transport of oxygen ($\overline{u'\,O_2'}$ in mol m$^{-2}$ d$^{-1}$) at 23° S for higBUS (b) and GR15 (d) of the upper 450 m as calculated with Eq. (17). Contour lines at -20., 0 , 20. in all panels.

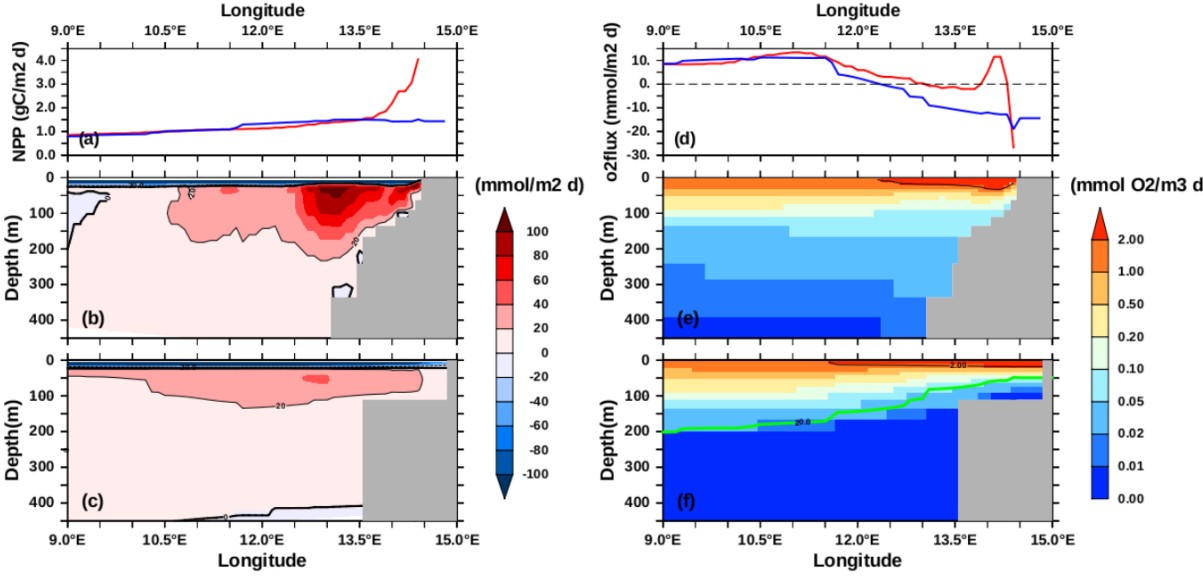

**Figure 17.** Climatological net primary production (a, in gC m$^{-2}$ d$^{-1}$) and climatological oxygen air-sea exchange (d, in mmol m$^{-2}$ d$^{-1}$) both at 23° S for higBUS (red) and GR15 (blue), zonal mean transport of detritus ($\overline{u\mathrm{Det}}$, in mmol P m$^{-2}$ d$^{-1}$) for higBUS (b) and GR15 (c), and O$_2$ consumption due to remineralization of detritus, DOC and grazer activity (e,f). Dashed line in (d) marks zero, positive values are O$_2$ flux into the air. Black contour lines in (b,c) are given for -20., 0 , 20. In panel (f), the green contour line refer to O$_2$ concentration at 20 mmol/m$^3$ to highlight the extended OMZ in GR15. In higBUS (e) all O$_2$ concentrations are well above this threshold (see also Fig. 11).





This indicates that, despite the high $O_2$ consumption fueled by high NPP and subsequent remineralization of organic matter, the physical $O_2$ supply compensates the biological demand and keeps the water column at a well ventilated level in higBUS.

Because all parameter settings of the biogeochemical model are identical in both set-up, we can attribute the emerging differences in the tracer distributions to the higher horizontal grid resolution of higBUS which allows for the development of meso-scale circulation structures. Our finding on the impact of horizontal model resolution is in line with earlier studies of

regional (e.g. Schmidt and Eggert (2016)) or global set-ups (Duteil et al., 2014; Buchanan and Tagliabue, 2021). Furthermore, the statement of Séférian et al. (2020) that biases found in the $O_2$ distribution of Earth system models are independent of the model resolution can not be supported by our results. The adequate representation of meso-scale dynamics is essential not only to achieve a good $O_2$ distribution (Buchanan and Tagliabue, 2021), but, moreover, to avoid the occurrence of unrealistic nitrogen losses. As we will show in the next sections, also temporal tracer trends are affected.





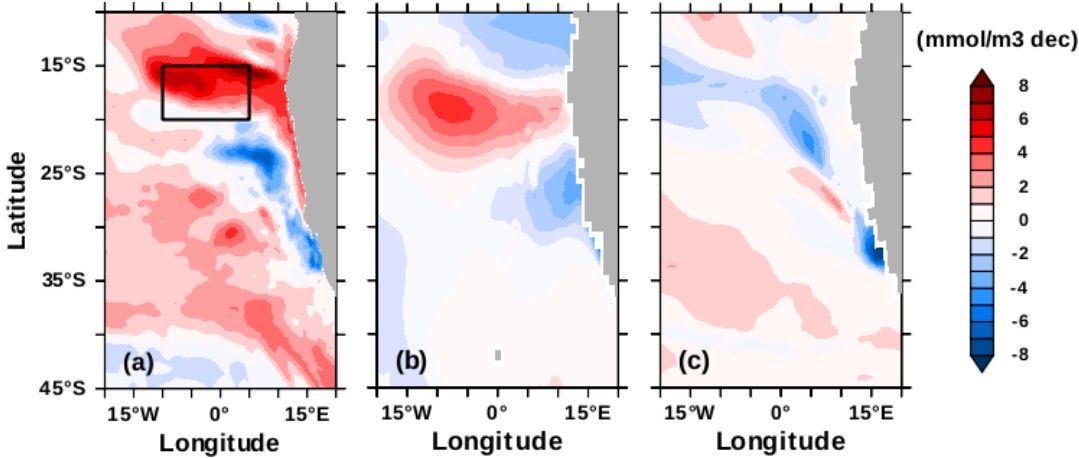

**Figure 18.** Mean oxygen trend (mmol m$^{-3}$ per decade) in 200-400 m over 1960-2009 for higBUS(a), deduced from observational data (Schmidtko et al., 2017) (b), and GR15 (c). The rectangle in (a) indicates the area for the O$_2$ budget given in Table 2.

**3.4   O$_2$ trends in the BUS region over 1960 to 2009**

The role of oxygen controlling the remineralisation pathways and, thus, the loss of bioavailable nitrogen gets addtional relevance in the context of climate change and ocean warming. Deoxygenation might enhance N-loss processes and could affect marine habitats with detrimental consequences of marine life (Schmidtko et al., 2017). A reduction of the global oceanic O$_2$ content of more than two per cent since the 1960s has been deduced (Schmidtko et al., 2017). The observed O$_2$ change is
not uniform, but eastern boundary upwelling regions are most vulnerable with a potential of an intensification of upwelling bringing more O$_2$ depleted water onto the shelf (Breitburg et al., 2018). The data compilation by Schmidtko et al. (2017) shows a dipole structure for the trend in O$_2$ of subsurface waters (200-400 m) in the BUS between 1960 and 2009 (Fig. 18b). In our simulation, a similar pattern is found in higBUS with slightly higher magnitudes and a small northward shift (Fig. 18a). The overall positive trend in the northern BUS with a maximum located south of 15° S reaches onto the shelf in higBUS and
extends southwards in a narrow coastal band. The observed trend does not show this small-scale feature on the shelf, which could be an artefact of limited data availability, especially in the last decade (see in Schmidtko et al. (2017) Extended Data Figure 5).

Between 25° S to 35° S, both, higBUS and the trend estimate from observations show a deoxygenation in the BUS. This decrease in O$_2$ is also present in other estimates of the decadal O$_2$ trend (Stramma et al., 2012; Ito et al., 2017). Interestingly,
in higBUS we find a positive decadal trend south of 30° S and west of 10° E which is missing in Schmidtko et al. (2017), but is also found in the data compilation of Stramma et al. (2012) (their Fig.2b). In our simulation, this positive trend pattern might be attributed the intensification of the wind stress curl over this region (Fig. A5). However, the poor data coverage, especially in the last decade 2000-2009 (Stramma et al., 2012; Schmidtko et al., 2017), creates a high uncertainty on the observed trend magnitude.





In GR15, we find a reversed and displaced dipole structure with smaller magnitude. The along-shore shelf areas north of 30°
S show hardly any trend over 1960-2009. As shown in Fig. 11c, $O_2$ concentrations of the shelf area are already completely
depleted which hampers a further decline.

    The similarity to the trend pattern from observations and highBUS, especially north of 30° S, allows us to gain insight into
the driving processes. Instead of looking at trends over the entire period 1960 to 2009 we calculate climatological means over

20 years of the periods 1960-1979 and 1990-2009. Anomalies are given as difference of the mean of 1990-2009 and the mean
of 1960-1979. We focus on the region 15° S to 20° S and 10° W to 5° E at 200-400 m with a positive $O_2$ trend (see box in Fig.
18a).

    Two mechanisms could provide additional oxygen at intermediate depth: 1) a decrease in $O_2$ consumption due to a reduced
NPP and, thus, reduced particle flux to the corresponding depth or 2) an increased ventilation of this depth horizon either by

vertical or lateral $O_2$ transport. NPP and particle fluxes into this box increase in both setups between the two time periods
(Table 2, Fig. A7). Consequentially, even more detritus is remineralized in the depth range of 200-400 m with an increase
of 13 % and 23 % of $O_2$ consumption due to remineralization for higBUS and GR15, respectively. The higher $O_2$ demand is
overcompensated by changes in lateral and vertical transport in higBUS, where the $O_2$ inventory of the box increases by more
than 15 %. In contrast, $O_2$ inventory in GR15 decreases slightly due to enhanced particle flux, with a concurrent small increase

in N-consumption due to remineralization (3 %) .

    In line with the higher absolute $O_2$ transport shown in Fig. 15, lateral transport components in higBUS are higher than in
GR15 (Table 2). The most striking difference between higBUS and GR15 is the vertical $O_2$ transport at 200 m and 400 m into
the box. While in higBUS an intense upwelling at 400 m carries oxygen into the box, the vertical transport at 200 m acts as
a small $O_2$ loss (upward transport). In GR15, the box is primarily ventilated from above (across 200 m), while small mean

upwelling velocities at the lower boundary play a minor role. However, the decreasing water mass age in higBUS over the
decades (Table 2) further indicates a supply of water being recently in contact with the atmosphere.

    In both set-ups, the net $O_2$ transport into the box increases over the decades. Because the chosen box covers a region with
diverging horizontal currents, the net zonal transport out of the box increases, but this is compensated by mean net meridional
and vertical transports into the box. The transport changes are a result of changes in the forcing which lead to a change in the

barotropic streamfunction (Fig. A6). The change in the barotropic streamfunction is also present in GR15 (Fig. A6d), but with
a different pattern and largest changes rather at the western side of the investigated area.

    The change in the dye tracers (Fig. A8) for the region of the box indicates that the additional oxygen comes primarily from
the Indian ocean (SIO) in higBUS. The intensification of the upwelling on the shelf between 10° S to 30° S and the off-shore
westward transport north at 20° S are clearly reflected in the change of the SIO dye tracer in higBUS, while this change is ab-

sent in GR15. The increased fraction of water from the SIO is linked to a stronger leakage of the Agulhas Current into the South
Atlantic at the end of the simulation period, which has also been identified by Biastoch et al. (2009). These authors attributed
the changed leakage strength to a relocation of southern hemisphere westerlies between 1970 and 2009. The $O_2$ transport for
the later period (1990-2009) and the decadal change of the $O_2$ transport are in line with a stronger Agulhas leakage (Fig. A9).
Both model resolutions show an enhanced $O_2$ transport with the Benguela current. However, we find that only in higBUS this





|  | higBUS | | GR15 | |
| --- | --- | --- | --- | --- |
|  | 1960-1979 | 1990-2009 | 1960-1979 | 1990-2009 |
| $O_2$ inventory ($10^{10}$ kmol) | 1.79 | 2.07 | 1.92 | 1.82 |
| zonal $O_2$ transport at 5° E (kmol/s) | 20.6 | 2.9 | 12.5 | 15.6 |
| zonal $O_2$ transport at 10° W (kmol/s) | -200.6 | -229.5 | -189.2 | -210.4 |
| meridional $O_2$ transport at 20° S (kmol/s) | **216.6** | **248.6** | **198.7** | **216.7** |
| meridional $O_2$ transport at 15° S (kmol/s) | 11.6 | **-5.7** | **-3.3** | **-11.6** |
| vertical $O_2$ transport across 200 m (kmol/s) | 1.9 | 0.2 | **-15.8** | **-23.8** |
| vertical $O_2$ transport across 400 m (kmol/s) | **18.5** | **19.9** | **4.3** | **1.0** |
| **total net $O_2$ transport into the box (kmol/s)** | **0.4** | **41.6** | **20.4** | **27.1** |
| column integrated NPP (kmol/s) | 806.9 | 833.7 | 907.3 | 1042.9 |
| org. matter particle flux at 100 m (kmol/s) | 122.5 | 129.7 | 137.4 | 167.6 |
| remineralization of detritus on $O_2$ (kmol/s) | 30.3 | 34.3 | 32.3 | 40.0 |
| remineralization of detritus on N (kmol/s) | 0.22 | 0 | 1.43 | 1.48 |
| mean temperature (°C) | 10.50 | 10.48 | 11.31 | 11.1 |
| mean age of water (yr) | 153.8 | 146.0 | 154.9 | 157.9 |

**Table 2.** Mean $O_2$ inventory of a box region (15-20° S and 10° W-5° E) over the 200-400 m depth range, zonal, meridional, and vertical mean $O_2$ transport across the box boundaries as well as the total net transport into the box, biological induced $O_2$ fluxes, and mean water mass properties. Positive numbers for transport indicate eastward (zonal), northward (meridional) or upward (vertical) fluxes. Bold numbers of transport are given to identify fluxes into the box more easily (e.g the negative vertical transport across 200 m in GR15 is an $O_2$ gain for the box). Note, that the particle flux is given at 100 m.

additional $O_2$ reaches the box region and compensates the $O_2$ demand from an increased NPP.

Within the model framework, we conclude that changes in $O_2$ in the northern BUS region, which are found in the observation over the last decades, are related to changes in the water mass composition with a higher contribution of SIO water. Changes in the wind stress forcing modify the upwelling pattern and lead to higher NPP along the coast. This trend pattern is only captured

by the high resolution model version. Our results underline the importance of a good representation of the physical conditions for the biogeochemistry. The coarse resolution setup is not suitable to depict the mean distribution of biogeochemical tracers and to resolve their trends in this eastern boundary upwelling region.




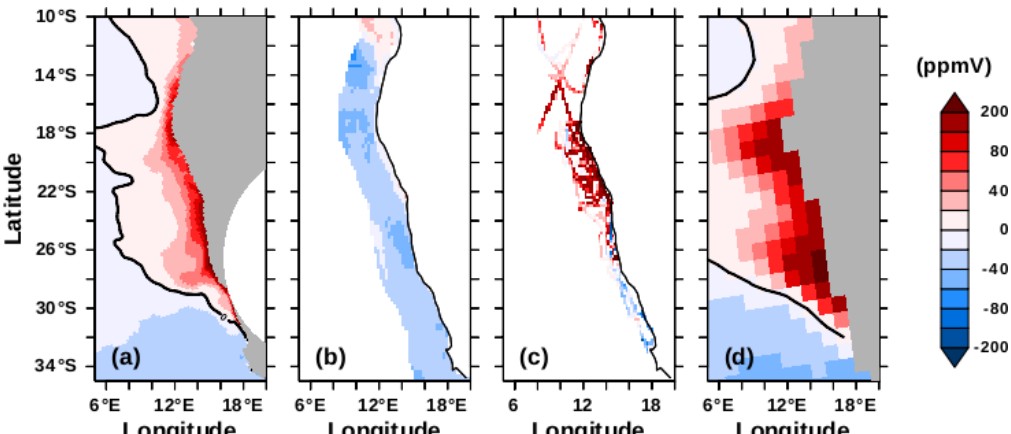

**Figure 19.** Annual mean gradient of pCO$_2$ between surface ocean and atmosphere ($\Delta$pCO$_2$ , ppmV) for higBUS(a), a two-step neural network method (b, calculated from Laruelle et al. (2018)), from several cruise tracks (c, Emeis et al. (2018)), for GR15 (d). Model data are averaged over 2000-2009. Contour line in (a) and (d) indicates 0.

## 4   Surface pCO$_2$ and decadal trends of the carbon inventory in the BUS

Recent publications raised the question, if the BUS region is rather a source or sink for atmospheric CO$_2$ (Emeis et al., 2018;
Laruelle et al., 2018). The partial pressure difference of CO$_2$ between the surface ocean and the atmosphere and, thus, the air-sea carbon fluxes are determined by the interplay between physical and biogeochemical conditions. With our setup we are able to investigate changes in the regional carbon cycle because we achieved a nearly steady state of our globally consistent model before we applied the transient ERA20C forcing. All horizontal and vertical tracer gradients in the global ocean are a result of the implemented physical and biogeochemical processes.

In higBUS, the entire region north of 31° S is a source for atmospheric CO$_2$ for the time period 2000-2009 (Fig. 19a). Upwelling of carbon rich water leads to outgassing despite the simulated high biological production rates which rather tend to reduce surface pCO$_2$. The comparison to GR15 underlines this impact of biological production. While physical parameters such as coastal upwelling (Fig. 8 and Fig. 10c, black line) and sea surface temperature of the adjacent open ocean (Fig. 9) are very similar in higBUS and GR15, the integrated NPP is higher in higBUS (Fig. 10). Thus, biological activity sequesters
dissolved inorganic carbon and exports it to greater depth and thereby reduces local outgassing of upwelled carbon rich water. The difference in NPP is related to the lack of bioavailable nitrogen in GR15, as has already been shown (Fig. 14), and which is a consequence of low oxygen conditions (see also Fig. A10 of the N/P ratio at 100 m). In higBUS, high $\Delta$pCO$_2$ values (> 20 ppmV, $\Delta$pCO$_2$ = pCO$_2$ in seawater minus pCO$_2$ in air) are found closer to the coast in a 400-500 km wide band. In GR15, higher positive $\Delta$pCO$_2$ values also stretch out far to the west which is not confirmed by measurements of previous pCO$_2$
compilations for the global open ocean (Takahashi et al., 2009; Bakker et al., 2016). South of 32° S both setups show a very similar small negative $\Delta$pCO$_2$ . The annual mean carbon flux is shown in the Appendix for completeness (Fig. A12).





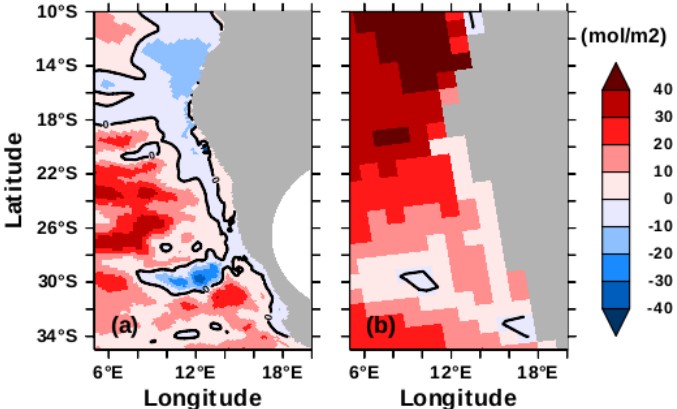

**Figure 20.** Difference of vertical integrated dissolved inorganic carbon concentration (mol/m$^2$) between 2009 and 1960 for higBUS (a) GR15
(b). Overlaid contour indicates 0.

Our result agrees well with a data collection of several ship cruises and volunteer observing ship lines by Emeis et al. (2018)
(Fig. 19c). They identified the entire BUS region north of 25° S as a carbon source for the atmosphere. Off the coast of Angola
and Namibia they found $\Delta pCO_2$ values of 150-200 ppmV. Fig. 19c shows the original data set sampled to a 0.25°x0.25° grid
(compare to Fig 13 in Emeis et al. (2018)). The southern part of the BUS is rather a sink region with negative $\Delta pCO_2$ according
to their data set. Still, one has to keep in mind that the data coverage over space and time is rather poor. A different approach
to estimate the surface $pCO_2$ in the BUS uses a two-step artificial neural network method to interpolate the $pCO_2$ data along
the continental margins with a spatial resolution of 0.25°x0.25° and with a monthly resolution for 1998 to 2015 (Laruelle
et al., 2018). The defined predictors of their method are particularly chosen for coastal conditions and applied to 10 different
biogeochemical provinces. This implies that their result is restricted to a maximal distance of 300 km off the coast or the 1000
m isobath. The gridded $pCO_2$ ocean data set refers nominal to the year 2006. Substracting a mean atmospheric $CO_2$ partital
pressure of 380.93 ppmV for 2006 (https://data.giss.nasa.gov/modelforce/ ghgases) results in a negative $\Delta pCO_2$ for the entire
BUS (Fig. 19b). In particular, the northern BUS contrasts with the findings of Emeis et al. (2018) and our model simulation.
Unfortunately, the data base of Laruelle et al. (2018) (compare their Fig. 2) is missing the near shore measurements of Emeis
et al. (2018). Therefore, the neural network might miss the observed extreme oceanic oversaturation.

However, neither observations nor the neural network give a convincing answer on the source/sink characteristic of the
BUS. From higBUS, we conclude that under climate conditions of the recent past the BUS is a weak carbon source for the
atmosphere with annual mean flux rates between 30-50 g m$^{-2}$ yr$^{-1}$ (Fig. A12).

To estimate the local anthropogenic carbon uptake of the BUS we calculate the change in the vertical integrated carbon
concentration between 2009 and 1960. Globally, both setup have a very similar C-uptake for 1901-2009 (higBUS: 96 PgC ;
GR15: 88.1 PgC, Fig. A13) which fits to estimates from literature (e.g. Sabine et al. (2004))

Despite the larger area with outgassing, GR15 shows an increase of the vertical integrated carbon inventory over period 1960-
2009 for the entire BUS region (Fig. 20). In contrast, in higBUS we see a decrease in the water column C-inventory, especially





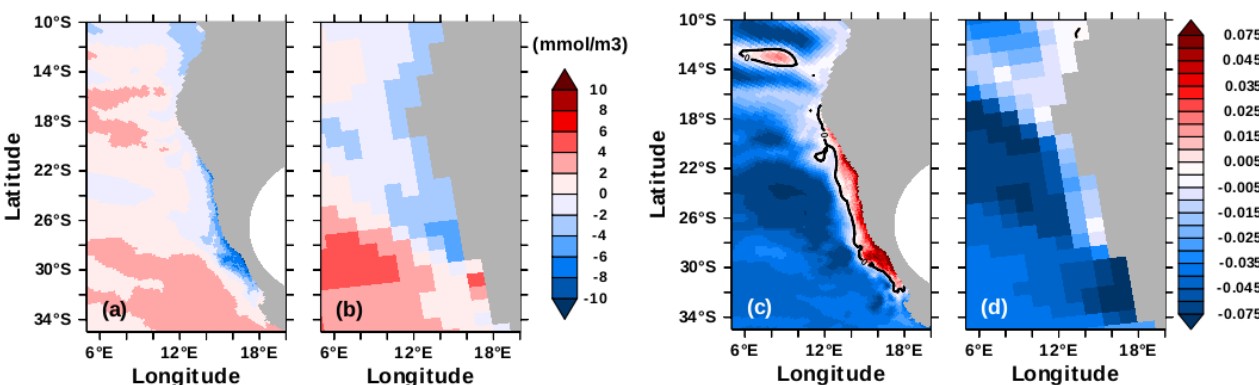

**Figure 21.** Difference between mean alkalinity (mmol/m$^3$) (a,b) and mean pH (c,d) of 1990-2009 and 1960-1979 averaged over the upper 0-150 m for higBUS (a,c) and GR15 (b,d)

along the coast of Africa, being a result of circulation changes and concurrent changes in the biochemistry. Alkalinity shows
only little change over the decades for both resolutions (Fig. 21), being primarily related to the interplay of an increased NPP and increased production of calcite shells south of 20° S.

Interestingly, coastal pH in the upper 150 m in higBUS even increases over the investigated period. Generally, upwelling areas are considered to be particularly vulnerable to ocean acidification (Gruber et al., 2012). In higBUS, coastal ocean acidification plays no role north of 32° S in the BUS where we find a small positive trend in pH, while GR15 shows a decreasing pH,
but at a smaller rate than estimated for the global ocean by Copernicus Marine Environmental Monitoring Service (-0.0016 yr$^{-1}$ for 1985-2018, https://marine.copernicus.eu/science-learning/ocean-monitoring-indicators). A recent study by Iida et al. (2021) estimated a global pH trend of -0.0018 yr$^{-1}$ for the surface ocean over 1993-2018. They also estimated pH changes for different regions and postulated lower change than expected from the atmospheric $CO_2$ trend in some regions, including the Angola-Benguela system. Thus, our simulated tendency of the temporal pH changes for the BUS is in line with estimates from
observations. However, the current data base on pH for BUS offers no possibility to evaluate our result for higBUS.



## 5 Summary and concluding remarks

In this study, we address the temporal evolution of the biogeochemistry in the Benguela upwelling system (BUS) over the 20th century. Therefore, we set up a global ocean biogeochemical model with a stretched grid configuration which allows also for resolving meso-scale circulation structures in the South Atlantic (higBUS). The advantages of this set-up are manifold:

a) it captures the Agulhas leakage as well as the impact of eddies which shed off the Agulhas current (Biastoch et al., 2009; Jungclaus et al., 2013) and carry oxygen rich water from the Indian Ocean into the BUS; b) it represents the complex frontal system and the poleward undercurrent of the BUS (Monteiro et al., 2011; Gutknecht et al., 2013); and c) its global approach avoids artefacts due to prescribed oceanic boundary conditions and allows for long and consistent model simulations, especially important in view of the marine carbon cycle. We set an additional focus on the development of the oxygen minimum zone

(OMZ) and nitrogen loss processes within the BUS. Therefore, we include a prognostic treatment of ammonium and nitrite and associated conversion processes like nitrification and anammox. The impact of meso-scale structures on the biogeochemistry is identified by running the same model in a coarser set-up ($1.5 \times 1.5°$, GR15) where the impact of eddies is parameterized with the common scheme from Gent et al. (1995).

    The evaluation of higBUS with climatological data, cruise data, and satellite derived products shows a high level of agree-

ment. In particular, the prognostic tracers $NO_2^-$ and $NH_4^+$ provide an additional assessment option for the role of anaerobic processes in OMZs. In addition, decadal oxygen trends are reproduced within the limit of observational uncertainty. In contrast, the coarser resolution set-up (GR15) has a very different performance. We find the development of an extended OMZ in the Benguela upwelling region with concurrent impacts on the biogeochemistry. Nitrogen loss processes are overestimated, resulting in a net primary production being limited by the lack of bioavailable nitrogen and unrealistically high concentrations

of $NO_2^-$ at intermediate depths. Most of these features are common to models typically used for climate change studies (horizontal resolution of 0.3 - 1.5 °, Séférian et al. (2020), Fig.3). Thus, our findings clearly show that model resolution matters. The impact of meso-scale structures, i.e. eddies and filaments, enhances vertical mixing and leads to a ventilation of the upper ocean. In higBUS, we find low $O_2$ (< 20 mmol/m$^3$) only on the shelf, which is in agreement with observations.

    Furthermore, the large scale circulation pattern and water mass properties in higBUS differ form those in GR15. The conse-

quences of a different mean circulation are evident in the multi-decadal trends of the oxygen. Only in higBUS, we find an increase in mid-depth $O_2$ being also reported in observational trend estimates (Schmidtko et al., 2017; Stramma et al., 2012; Ito et al., 2017). The results of higBUS are in agreement with model studies of age and pathways of the water masses in BUS of Tim et al. (2018) who applied water parcel tracking methods over the period 1960-2009. They speculated on biogeochemical changes such as increased oxygen supply of the BUS induced by the intensification of the Agulhas leakage which is supported

by our findings from higBUS.

    All these resolution dependent differences eventually affect the carbon cycle and the water column carbon inventory change over the 20th century. In principle, we find for both model resolutions that the northern BUS is a carbon source for the atmosphere, while the southern BUS is rather a sink. However, the source region in GR15 is much more extended into the South Atlantic, which disagrees, at least for the region south of 22° S, with the observational estimates (Takahashi et al.,





2009; Bakker et al., 2016). More striking is the temporal change of the water column carbon inventory for higBUS and GR15. Despite a very similar global uptake of anthropogenic carbon in both set-ups, the regional C-inventory increase over the entire simulation is smaller in higBUS. For the last 5 decades (1960-2009) higBUS projects a negative trend close to the coast.

The different trends of the carbon chemistry in higBUS and GR15 also show up in the local pH trend. In higBUS, pH changes on the shelf north of 32 ° S are even slightly positive. Thus, according to our model, there is no impact of ocean acidification 625 over the last decades in the BUS. Unfortunately, there is no long term data set on pH for this region to evaluate our finding.

Despite the better performance of the set-up with meso-scale structures, we have to consider that the higBUS experiment has several shortcomings: (1) The spatial resolution of the chosen forcing from reanalysis data (ERA20C) is rather coarse (1 degree) which might have the tendency to underestimate coastal upwelling as discussed by Milinski et al. (2016). Especially, a wind product with higher resolution, i.e. QuikSCAT and ASCAT with 0.25° resolution, might enhance the upwelling strength 630 (Junker et al., 2015). Unfortunately, these higher resolved data products cover only one decade and are, thus, not suitable for our purpose. (2) The biogeochemical model includes a comprehensive representation of the nitrogen cycle, but it does not include remineralization processes based on the activity of chemolithoautothrophic sulphur bacteria (see e.g. Schmidt and Eggert (2016)). They are active in anoxic sediments layers and could supply ammonium to the bottom layer from DNRA (dissimilatory nitrate reduction to ammonium). (3) Furthermore, our setup does not include migrating zooplankton. Zooplankton degrades 635 organic material and consumes $O_2$. Migration could therefore have an impact on the oxygen distribution in the water column. However, we are not convinced that the behavior of migrating zooplankton in the vicinity of OMZs is well enough understood, yet (Escribano et al., 2009).

In summary, our results emphasize that it is crucial to resolve meso-scale circulation structures to achieve a reasonable oxygen distribution and, concurrently, a good representation of the nutrient cycle in the BUS. The coarse resolution version, 640 GR15, suffers from too little mixing and too low exchange between the open ocean and the shelf areas. The resulting extended OMZ affects not only the nitrate reduction processes, but also the local primary production and, thus, the biogeochemistry of the BUS. By simulating a comprehensive nitrogen cycle these weaknesses of the coarse resolution set-up clearly shows up in the pattern of $NO_2^-$. Thus, in view of a potentially increasing number of $NO_2^-$ measurements, a comprehensive nitrogen cycle could be a promising validation tool for biogeochemical models. In addition, we found that the impact of meso-scale 645 structures also changes the transient behavior of the local $O_2$ distribution, the regional $CO_2$ uptake, and the temporal change of sea water pH. Therefore, we are convinced that, at least for some regions, the projected future pathway of changes in the ocean biogeochemistry cannot be reliably derived from coarse resolution models.

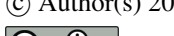



*Code and data availability.*   Primary data and code for this study are available from the corresponding author.





## Appendix A: Supplementary Information

### A1    Parameters of the new nitrogen cycle

| Variable | Symbol | Value | Units | References |
|----------|--------|-------|-------|------------|
| half sat. const. of nitrate for phytoplankton | $bk_{\mathrm{NO_3}}$ | 0.16 | mmol m$^{-3}$ | 0.3-1.5 (a,g) |
| half sat. const. of ammonium for phytoplankton | $bk_{\mathrm{NH_4}}$ | 0.1 | mmol m$^{-3}$ | 0.1-0.3 (b,g) |
| half sat. const. of nitrate for diazotrophs | $bk_{\mathrm{NO_3}}$ | 0.016 | mmol m$^{-3}$ | (h) |
| half sat. const. of ammonium diazotrophs | $bk_{\mathrm{NH_4}}$ | 0.01 | mmol m$^{-3}$ | (h) |
| max. oxygen concentration for nitrogen reduction processes | $\sigma_{crit}$ | 20 | mmol m$^{-3}$ | (e) |
| rate of $DNRN$ | $\lambda_{dnrn}$ | 0.15 | d$^{-1}$ | 0.2 (g) |
| half sat. const. of $DNRN$ | $bk_{dnrn}$ | 0.05 | mmol m$^{-3}$ | (g) |
| rate of $DNRA$ | $\lambda_{dnra}$ | 0.1 | d$^{-1}$ | 0.05 (g) |
| half sat. const. of $DNRA$ | $bk_{dnra}$ | 0.5 | mmol m$^{-3}$ | (g) |
| rate of $NRN2$ | $\lambda_{nrn2}$ | 0.008 | d$^{-1}$ | (g) |
| half sat. const. of $NRN2$ | $bk_{nrn2}$ | 0.1 | mmol m$^{-3}$ | 0.5 (g) |
| rate of $Anam$ | $\lambda_{anam}$ | 0.005 | d$^{-1}$ | 0.05 (d) |
| NO$_3^-$ production fraction during $Anam$ | $\epsilon_a$ | 0.3 | | (i) |
| max. nitrification rate of NH$_4^+$ | $\lambda_{oxy\mathrm{NH_4}}$ | 0.17 | d$^{-1}$ | (d,f) |
| max. nitrification rate of NO$_2^-$ | $\lambda_{oxy\mathrm{NO_2}}$ | 0.17 | d$^{-1}$ | (f) |
| light inhibition const. for nitrification | $bk_I$ | 0.1 | W m$^{-2}$ | (c) |

**Table A1.** Parameter values used in Eq.(1) - (13) of the newly implemented nitrogen cycle and literature values, if available. If a range is given, we adjusted the number, otherwise we take the value from the given reference. (a) Lima and Doney (2004), (b) Vallina and Quéré (2008), (c) Fennel et al. (2006), (d) Gutknecht et al. (2013) and refs within, (e) Kalvelage et al. (2013), (f) Azhar et al. (2014), (g) Beckmann and Hense (2017), (h) Paulsen et al. (2017), (i) Brunner et al. (2013)





## A2    Used observational cruise data sets


We use data from four GENUS cruises:

| Campaigns | Cruise dates | References |
| --- | --- | --- |
| Africana 258 | 01.12.2009 - 18.12.2009 | https://doi.pangaea.de/10.1594/PANGAEA.880406 |
| Discovery | 10.09.2010 - 13.10.2010 | https://doi.pangaea.de/10.1594/PANGAEA.779134 |
| Maria S. Merian 17 | 30.01.2011 - 07. 02.2011 | https://doi.org/10.1594/PANGAEA.843443 |
| Meteor 103/1 | 28.12.2013 - 10.02.2014 | https://doi.pangaea.de/10.1594/PANGAEA.854031 |

**Table A2.** Expedition names and dates, including links to the data sets in the PANGAEA World Data Center.

## A3    Climatological pattern of water mass composition

Fig. A1 shows the water mass composition at 30° S for higBUS and GR15 for water from 6 regions. Water from SIO has the largest contribution on the shelf and off shore. Only the dye tracer (sBUS), which origins in the area that hosts this section, has

higher values in the upper water column. Water coming from SATL contributes to 15-20 % to shelf water (<400 m depth). In higBUS, there is also some fraction ( < 20 %) of water from SACW reaching 30° S on the shelf.

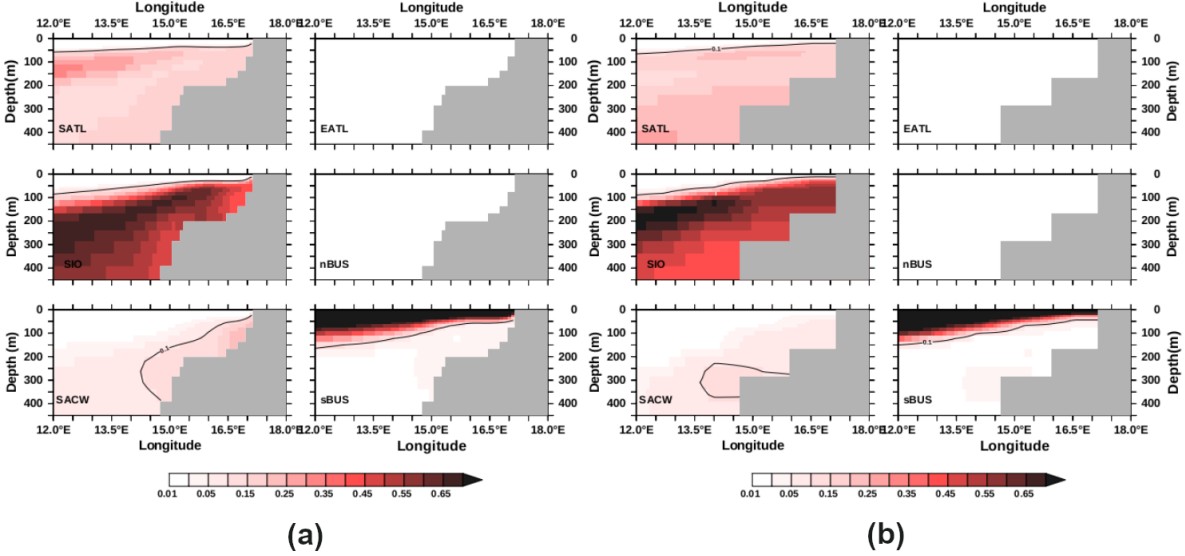

**Figure A1.** Climatological water mass fraction from 6 regions, SATL, SIO, SACW, EATL, nBUS, and sBUS, along 30° S for higBUS (a) and GR15 (b). Contour line at 0.1. See Fig. 3 for location of dye source regions. Water masses from EATL and nBUS are not found at this latitude (fraction <0.01)



Fig. A2 shows the water mass composition at 23° S for higBUS and GR15 for water from 6 regions. As shown in Fig. 6, water from the SIO contributes the largest fraction west of 12° E below 150 m. This is a result of the transport by the Benguela current which is clearly identifiable in higBUS, but has a more diffusive structure in GR15. About 20 % of the water mass
origins from SATL, being mixed in the Benguela current west of the Cape Basin (Fig. A1) in both resolutions. In higBUS, the shelf (< 400 m) is primarily occupied by water from SACW with additional small contributions of water from the equatorial Atlantic (EATL, 5 %) and nBUS (>10 %) at this northern BUS section . The dye tracer, that origins in the BUS part which also hosts this section (sBUS), shows, of course, the highest fraction in the upper 100 m of the water column. However, the more efficient upwelling and the concurrent stronger westward transport in higBUS is reflected in the pattern of this dye tracer,
which is confinded to the upper 80 m of water column on the shelf. In contrast, in GR15 we find a more diffusive picture with a higher fraction of this dye tracers found in the bottom water on the shelf.

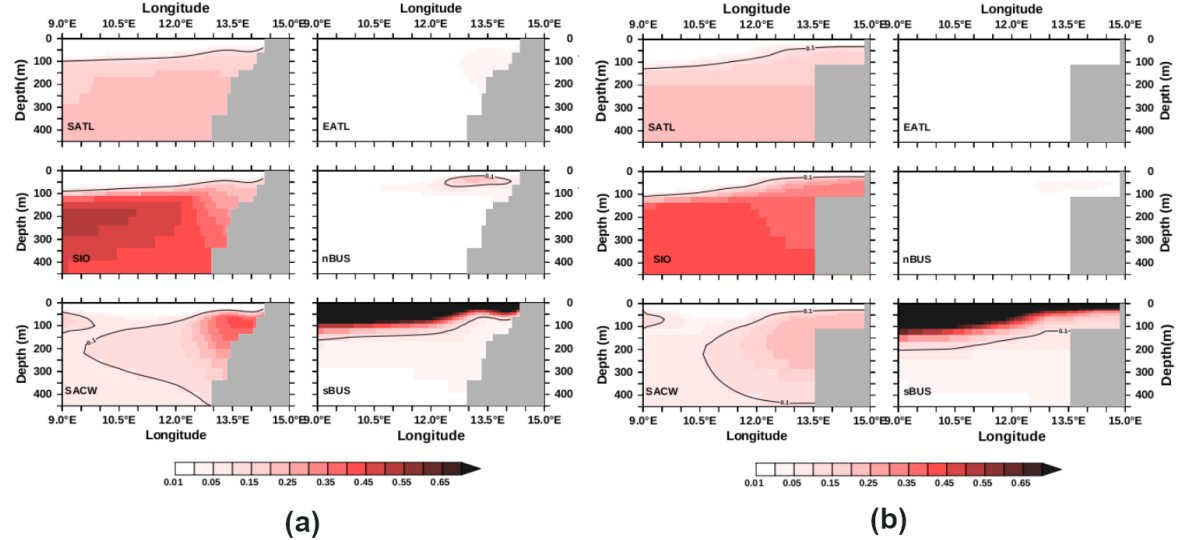

**Figure A2.** Climatological water mass fraction from 6 regions, SATL, SIO, SACW, EATL, nBUS, and sBUS, along 23° S for higBUS (a) and GR15 (b). See Fig. 3 for location of dye source regions.





## A4 Presence of eddy structures in higBUS

The impact of the higher horizontal resolution on the flow field becomes visible by the sea surface height (SSH) anomaly, here shown for the southern BUS (Fig. A3). In this daily mean snapshot for April, 10th 2009 higBUS shows meso-scale eddy
structures of 100 km diameter which are travelling westward into the Atlantic. The clear displacement of the anticyclonic eddies is highlighted by the 3 cm contour line of the SSH-anomaly also shown for 10 and 20 days after April, 10th. In GR15, we see a much broader positive SSH anomaly which is rather stationary and fully flattens within 20 days (see only one contour line in Fig. A3b). As seen for surface (Fig. 7a), subsurface (Fig. 5a) and upwelling velocities (Fig. 8) higBUS also shows a SSH anomaly pattern with higher variability indicating a vivid mixing of water masses, especially south of 30° S.

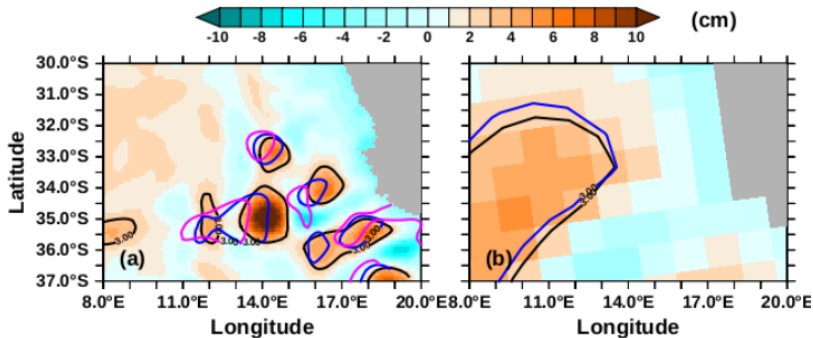

**Figure A3.** Sea surface height (SSH) anomaly (cm) on April, 10th 2009 minus a 30 days running mean of SSH of 2009 for higBUS(a) and GR15(b). Overlaid are contours of 3 cm SSH anomaly for April 10th (black), 20th (blue), and 30th (purple) to indicated the eddy propagation.

## A5 Climatological mixed layer depth

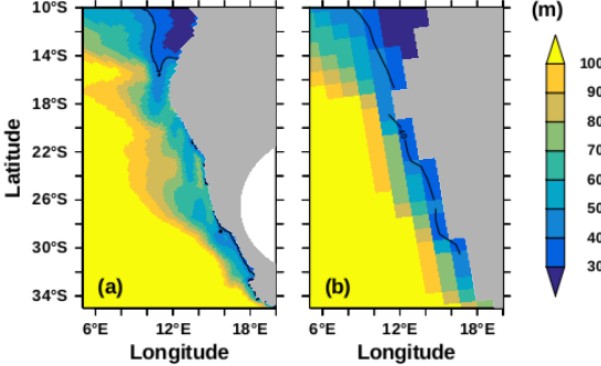

**Figure A4.** Climatological mean of the annual maximum of the monthly mean mixed layer depth (m). Contour line is given for 40 m.





## A6    Temporal changes of the wind stress forcing and barotropic streamfunction between 1960 and 2009

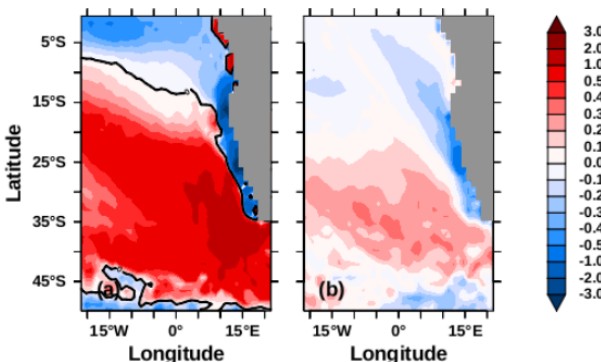

**Figure A5.** Mean wind stress curl ($10^{-4}$ N m$^{-2}$ km$^{-1}$) in ERA20C for 1960-1979 (a) and the anomaly of (1990-2009) - (1960-1979) (b). Note non-linear color scale.

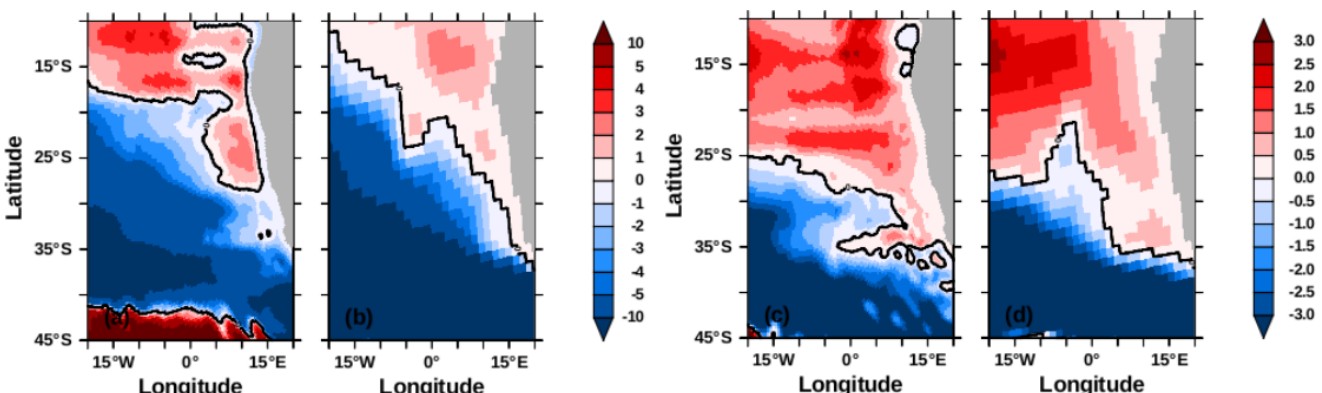

**Figure A6.** Mean barotropic streamfunction (Sv) of 1960-1979 for higBUS (a) and GR15 (b) and change of the mean barotropic streamfunction between 1990-2009 and 1960-1979 for higBUS (c) and GR15 (d).





## A7  Temporal change of the NPP and the water mass distribution

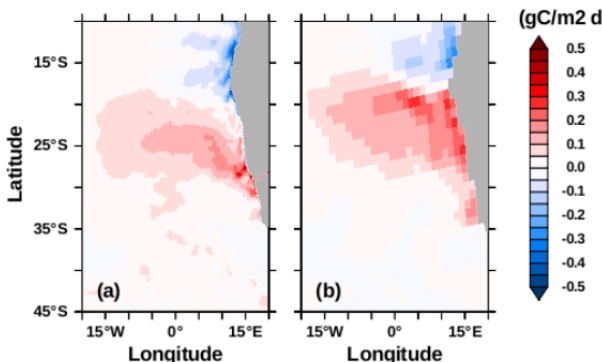

**Figure A7.** Temporal change in NPP (gC m$^{-2}$ d$^{-1}$) between (1990-2009) and (1960-1979) for higBUS (c) and GR15 (d).

As shown for the zonal section of 23° S in Fig. A2, water with the origin from SATL contributes 20-25 % to the water mass composition of the BUS in the northern part (north of 30° S) while the southern BUS gets less than 10 % of this water at a mean depth of 100-200 m in higBUS (Fig. A8). Water from SACW arrives with the shelf-bound poleward current into the BUS. Therefore higher fractions are found only on the shelf (Fig. A2) in higBUS. The temporal change of both dye tracers, SATL and SACW, is different from the change we found in O$_2$ which underlines the finding that it is primarily the change of the Agulhas leakage that is responsible for the O$_2$ anomaly north of 20° S. The large scale patterns of both tracers in GR15 have some similarity to higBUS with the already discussed deviations, but the temporal change is different. Particularly, water coming from the SATL has a slightly higher impact on coastal regions south of 30° S in the later period 1990-2009. This might explain the increased NPP south of 30° S (+20 %, Fig. A7) which causes also a pronounced O$_2$ change not being observed at that magnitude.

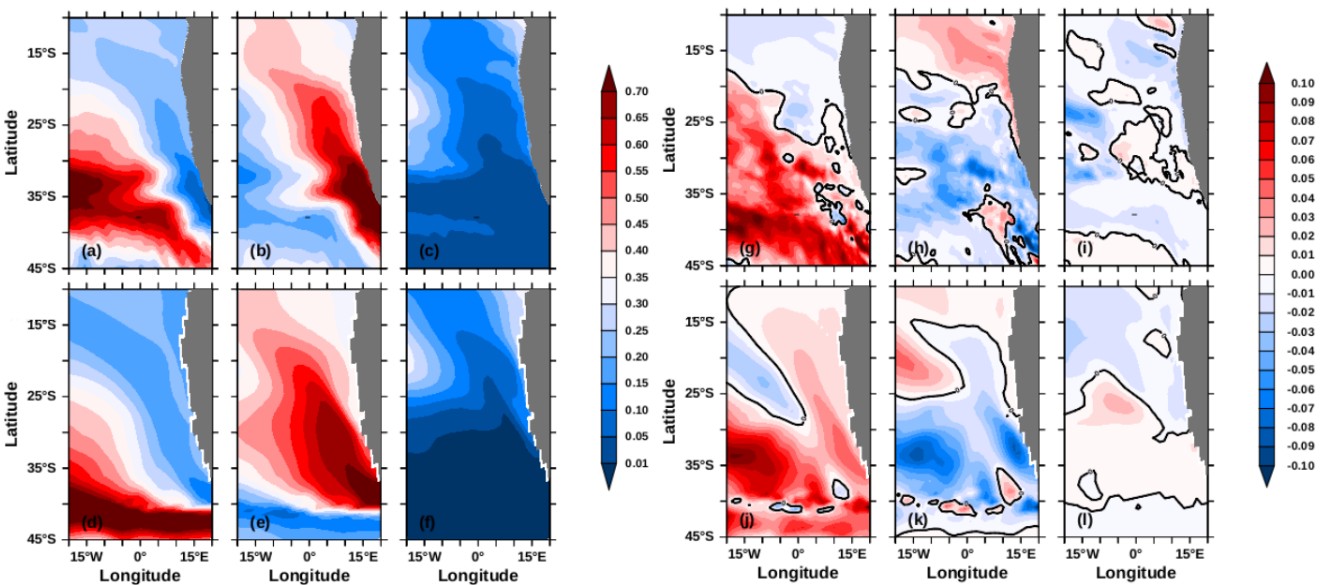

**Figure A8.** Mean water mass fraction (1960-1979) at 200-400 m from the South Atlantic (SATL, a,d), the southern Indian Ocean (SIO,b,e), and the South Atlantic Central Water (SACW, c,f) for higBUS (a,b,c) and GR15 (d,e,f). Anomaly of the water mass fraction (1990-2009)-(1960-1979) for the same dye tracers: SATL (g,j), SIO (h,k), and SACW (i,l) for higBUS (g,h,i) and GR15 (j,k,l). Contour line is given at 0.

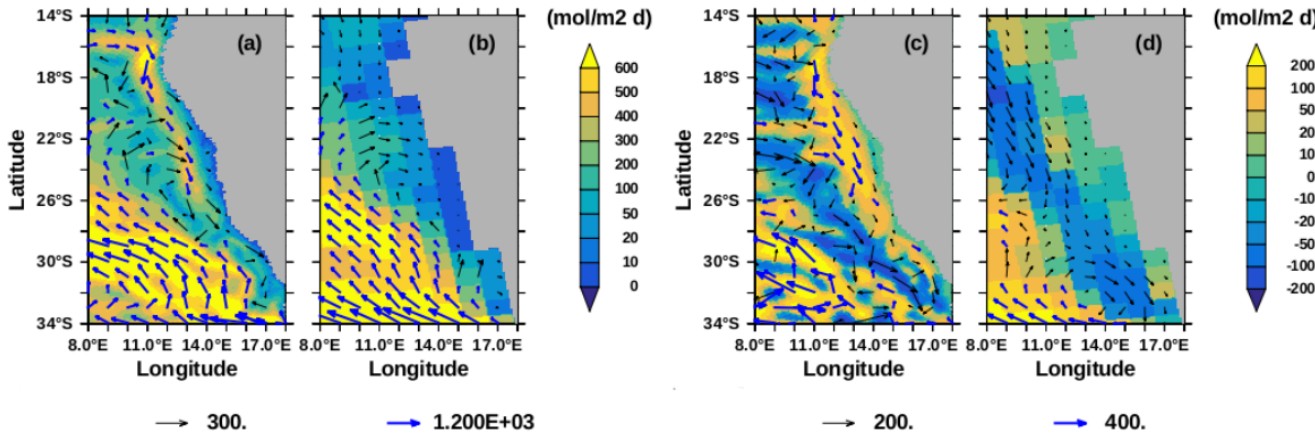

**Figure A9.** As Fig. 15, but absolute mean lateral $O_2$ transport averaged over 1990-2009 (a,b) and the difference of the absolute mean lateral $O_2$ transport between (1990-2009) and (1960-1979) (c,d) for higBUS (a,c) and GR15 (b,d).





## A8   N/P ratio and climatological remineralization of detritus

The comparison of the N/P ratio at 100 m clearly shows the depleted coastal area in GR15 (Fig. A10). There are some
discrepancies in the pattern between higBUS and the WOA data. However, the ratio is very sensitive to quality of phosphate
measurements. Another N/P estimates from data miss e.g. the very low ratio west of 12° E (Galbraith and Martiny, 2015).

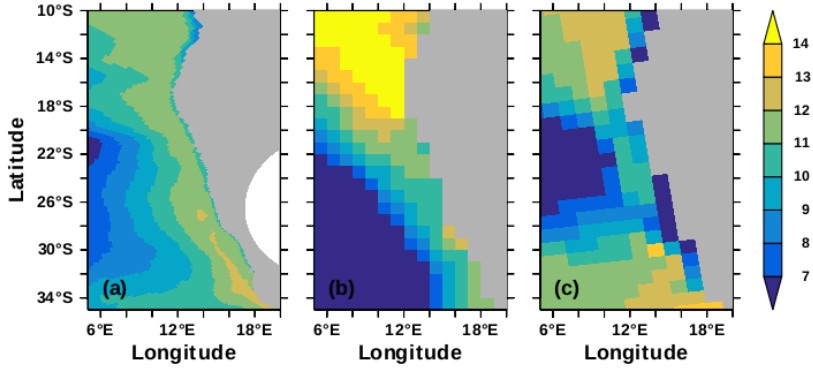

**Figure A10.** Mean nitrate to phosphate ratio for 50-100 m for higBUS(a), from WOA data (b), and GR15(c). Model results are means over
1960-2009.

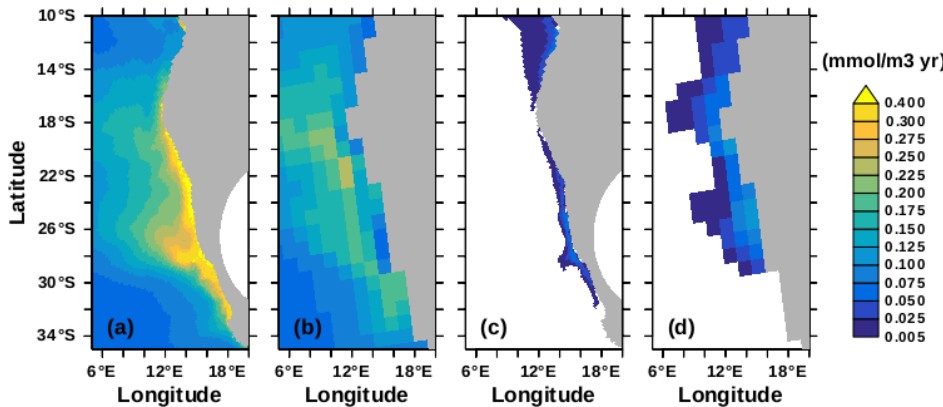

**Figure A11.** Climatological mean remineralization of organic material in the upper 200 m (mmol P m$^{-3}$yr$^{-1}$ ) based on dissolved oxygen
(a,b) or N-components(c,d) for higBUS(a,c) and GR15(b,d).




## A9 Climatological carbon flux and anthropogenic carbon uptake

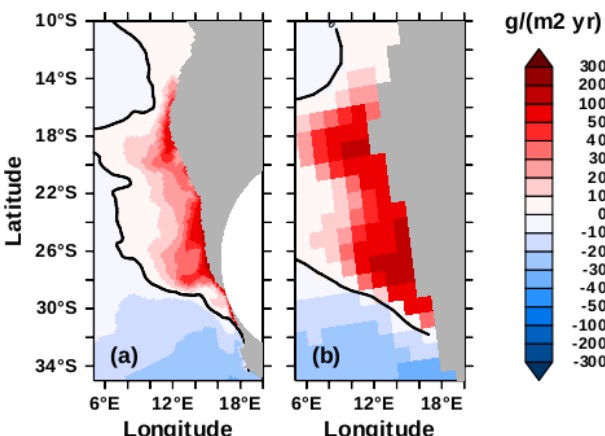

**Figure A12.** Annual mean carbon flux between surface ocean and atmosphere (g m$^{-2}$ yr$^{-1}$) for 2000-2009: higBUS(a), GR15 (b). Positive values denote fluxes to the atmosphere. Note non-linear color scale. Contour line indicates 0.

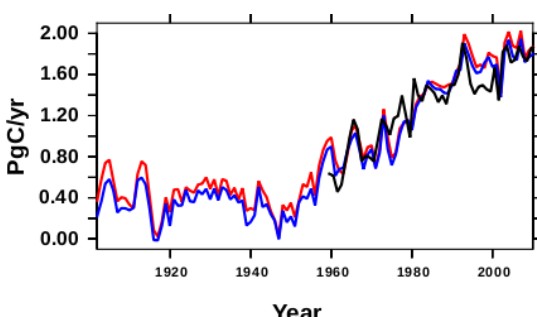

**Figure A13.** Annual mean global net carbon flux (Pg/yr) into the ocean from 1901 to 2009 for higBUS (red) and GR15 (blue). Overlaid is the result of a simulation with GR15 as a member of the Global Carbon Budget (black, Quéré et al. (2015) being forced with NCEP reanalysis data, 1958-2009)

*Author contributions.* Katharina D. Six developed and tuned the nitrogen cycle for HAMOCC and set up the stretched grid configuration of higBUS. She perfomed and analysed all simulations. Both authors critically discussed the presented results and contributed by providing valuable feedback during the manuscript compilation.

*Competing interests.* The authors declare that they have no conflict of interest.






*Acknowledgements.* All simulations were performed at the German Climate Computing Center (DKRZ). Katharina D. Six was supported through the Cluster of Excellence 'CliSAP' (EXC177), Universität Hamburg, funded through the German Research Foundation (DFG). We thank Tim Rixen for provding the data from the Genus cruises and the surface pCO2 data set from Emeis et al. (2018). We also thank M. Mathis for his internal review of the manuscript.



## Appendix: References

Aumont, O., Orr, J. C., Monfray, P., Madec, G., and Maier-Reimer, E.: Nutrient trapping in the equatorial Pacific: The ocean circulation solution, Global Biogeochemical Cycles, 13, 351–369, https://doi.org/https://doi.org/10.1029/1998GB900012, 1999.

Azhar, M. A., Canfield, D. E., Fennel, K., Thamdrup, B., and Bjerrum, C. J.: A model-based insight into the coupling of nitrogen and sulfur cycles in a coastal upwelling system, Journal of Geophysical Research: Biogeosciences, 119, 264–285, https://doi.org/10.1002/2012jg002271, 2014.

Bakker, D. C. E., Pfeil, B., Landa, C. S., Metzl, N., O'Brien, K. M., Olsen, A., Smith, K., Cosca, C., Harasawa, S., Jones, S. D., Nakaoka, S., Nojiri, Y., Schuster, U., Steinhoff, T., Sweeney, C., Takahashi, T., Tilbrook, B., Wada, C., Wanninkhof, R., Alin, S. R., Balestrini, C. F.,

Barbero, L., Bates, N. R., Bianchi, A. A., Bonou, F., Boutin, J., Bozec, Y., Burger, E. F., Cai, W.-J., Castle, R. D., Chen, L., Chierici, M., Currie, K., Evans, W., Featherstone, C., Feely, R. A., Fransson, A., Goyet, C., Greenwood, N., Gregor, L., Hankin, S., Hardman-Mountford, N. J., Harlay, J., Hauck, J., Hoppema, M., Humphreys, M. P., Hunt, C. W., Huss, B., Ibánhez, J. S. P., Johannessen, T., Keeling, R., Kitidis, V., Körtzinger, A., Kozyr, A., Krasakopoulou, E., Kuwata, A., Landschützer, P., Lauvset, S. K., Lefèvre, N., Lo Monaco, C., Manke, A., Mathis, J. T., Merlivat, L., Millero, F. J., Monteiro, P. M. S., Munro, D. R., Murata, A., Newberger, T., Omar, A. M., Ono, T., Paterson,

K., Pearce, D., Pierrot, D., Robbins, L. L., Saito, S., Salisbury, J., Schlitzer, R., Schneider, B., Schweitzer, R., Sieger, R., Skjelvan, I., Sullivan, K. F., Sutherland, S. C., Sutton, A. J., Tadokoro, K., Telszewski, M., Tuma, M., van Heuven, S. M. A. C., Vandemark, D., Ward, B., Watson, A. J., and Xu, S.: A multi-decade record of high-quality $f\mathrm{CO_2}$ data in version 3 of the Surface Ocean $\mathrm{CO_2}$ Atlas (SOCAT), Earth System Science Data, 8, 383–413, https://doi.org/10.5194/essd-8-383-2016, 2016.

Beckmann, A. and Hense, I.: The impact of primary and export production on the formation of the secondary nitrite maximum: A model
study, Ecological Modelling, 359, 25–33, https://doi.org/10.1016/j.ecolmodel.2017.05.014, 2017.

Biastoch, A., Böning, C. W., Schwarzkopf, F. U., and Lutjeharms, J. R. E.: Increase in Agulhas leakage due to poleward shift of Southern Hemisphere westerlies, Nature, 462, 495–498, https://doi.org/10.1038/nature08519, 2009.

Bopp, L., Resplandy, L., Orr, J., Doney, S., Dunne, J., Gehlen, M., Halloran, P., Heinze, C., Ilyina, T., Séférian, R., Tjiputra, J., and Vichi, M.: Multiple stressors of ocean ecosystems in the 21st century: projections with CMIP5 models, Biogeosciences,
https://doi.org/doi:10.5194/bg-10-6225-2013, 2013.

Boyer, Tim P.and Garcia, H. E., Locarnini, R. A., Zweng, M. M., Mishonov, A. V., Reagan, J. R., Weathers, K. A., Baranova, O. K., Seidov, D., and Smolyar, I. V.: World Ocean Atlas 2018, 2018.

Breitburg, D., Levin, L. A., Oschlies, A., Grégoire, M., Chavez, F. P., Conley, D. J., Garçon, V., Gilbert, D., Gutiérrez, D., Isensee, K., Jacinto, G. S., Limburg, K. E., Montes, I., Naqvi, S. W. A., Pitcher, G. C., Rabalais, N. N., Roman, M. R., Rose, K. A.,
Seibel, B. A., Telszewski, M., Yasuhara, M., and Zhang, J.: Declining oxygen in the global ocean and coastal waters, Science, 359, https://doi.org/10.1126/science.aam7240, 2018.

Brunner, B., Contreras, S., Lehmann, M. F., Matantseva, O., Rollog, M., Kalvelage, T., Klockgether, G., Lavik, G., Jetten, M. S. M., Kartal, B., and Kuypers, M. M. M.: Nitrogen isotope effects induced by anammox bacteria, Proceedings of the National Academy of Sciences, 110, 18 994–18 999, https://doi.org/10.1073/pnas.1310488110, 2013.

Buchanan, P. J. and Tagliabue, A.: The Regional Importance of Oxygen Demand and Supply for Historical Ocean Oxygen Trends, Geophysical Research Letters, 48, https://doi.org/10.1029/2021gl094797, 2021.

Cabré, A., Marinov, I., Bernardello, R., and Bianchi, D.: Oxygen minimum zones in the tropical Pacific across CMIP5 models: mean state differences and climate change trends, Biogeosciences, 12, 5429–5454, https://doi.org/10.5194/bg-12-5429-2015, 2015.



Chelton, D. B., deSzoeke, R. A., Schlax, M. G., El Naggar, K., and Siwertz, N.: Geographical Variability of the First

Baroclinic Rossby Radius of Deformation, Journal of Physical Oceanography, 28, 433–460, https://doi.org/10.1175/1520-0485(1998)028<0433:GVOTFB>2.0.CO;2, 1998.

Cocco, V., Joos, F., Steinacher, M., Frölicher, T. L., Bopp, L., Dunne, J., Gehlen, M., Heinze, C., Orr, J., Oschlies, A., Schneider, B., Segschneider, J., , and Tjiputra, J.: Oxygen and indicators of stress for marine life in multi-model global warming projections, Biogeosciences, https://doi.org/doi: 10.5194/bg-10-1849-2013, 2013.

Cunningham, S. A., Alderson, S. G., King, B. A., and Brandon, M. A.: Transport and variability of the Antarctic Circumpolar Current in Drake Passage, Journal of Geophysical Research: Oceans, https://doi.org/doi:10.1029/2001JC001147, 2003.

Dee, D. P., Uppala, S. M., Simmons, A. J., Berrisford, P., Poli, P., Kobayashi, S., Andrae, U., Balmaseda, M. A., Balsamo, G., Bauer, P., Bechtold, P., Beljaars, A. C. M., van de Berg, L., Bidlot, J., Bormann, N., Delsol, C., Dragani, R., Fuentes, M., Geer, A. J., Haimberger, L., Healy, S. B., Hersbach, H., Hólm, E. V., Isaksen, L., Kållberg, P., Köhler, M., Matricardi, M., McNally, A. P., Monge-Sanz,

B. M., Morcrette, J.-J., Park, B.-K., Peubey, C., de Rosnay, P., Tavolato, C., Thépaut, J.-N., and Vitart, F.: The ERA-Interim reanalysis: configuration and performance of the data assimilation system, Quarterly Journal of the Royal Meteorological Society, 137, 553–597, https://doi.org/10.1002/qj.828, 2011.

Desbiolles, F., Blanke, B., and Bentamy, A.: Short-term upwelling events at the western African coast related to synoptic atmospheric structures as derived from satellite observations, J. Geophy. Res., https://doi.org/doi:10.1002/2013JC009278, 2014.

Dietze, H. and Loeptien, U.: Revisiting "nutrient trapping" in global coupled biogeochemical ocean circulation models, Global Biogeochemical Cycles, 27, 265–284, https://doi.org/https://doi.org/10.1002/gbc.20029, 2013.

DiMarco, S. F., Chapman, P., Nowlin Jr, W. D., Hacker, P., Donohue, K., Luther, M., Johnson, G. C., and Toole, J.: Volume transport and property distribution of the Mozambique Channel, Deep-Sea Res. II, 2002.

Duteil, O., Schwarzkopf, F. U., Böning, C. W., and Oschlies, A.: Major role of the equatorial current system in setting oxy-

gen levels in the eastern tropical Atlantic Ocean: A high-resolution model study, Geophysical Research Letters, 41, 2033–2040, https://doi.org/https://doi.org/10.1002/2013GL058888, 2014.

Emeis, K., Eggert, A., Flohr, A., Lahajnar, N., Nausch, G., Neumann, A., Rixen, T., Schmidt, M., Van der Plas, A., and Wasmund, N.: Biogeochemical processes and turnover rates in the Northern Benguela Upwelling System, Journal of Marine Systems, p. 63–80, 2018.

Eppley, R. W.: Temperature and phytoplankton growth in the sea, Fish. Bull., 1972.

Escribano, R., Hidalgo, P., and Krautz, C.: Zooplankton associated with the oxygen minimum zone system in the northern upwelling region of Chile during March 2000, Deep Sea Research Part II: Topical Studies in Oceanography, 56, 1083–1094, https://doi.org/10.1016/j.dsr2.2008.09.009, 2009.

Espinoza-Morriberón, D., Echevin, V., Gutiérrez, D., Tam, J., Graco, M., Ledesma, J., and Colas, F.: Evidences and drivers of ocean deoxygenation off Peru over recent past decades, Scientific Reports, 11, https://doi.org/10.1038/s41598-021-99876-8, 2021.

ESR: OSCAR third degree resolution ocean surface currents, Ver. 1. PO.DAAC, CA, USA., Dataset accessed 2020-04-17 at, https://doi.org/10.5067/OSCAR-03D01, 2009.

Fennel, K.: Theory of the Benguela Upwelling System, Journal of Phys. Oce., pp. 177–190, 1998.

Fennel, K., Wilkin, J., Levin, J., Moisan, J., O'Reilly, J., and Haidvogel, D.: Nitrogen cycling in the Middle Atlantic Bight: Results from a three-dimensional model and implications for the North Atlantic nitrogen budget, Global Biogeochemical Cycles, 20,

https://doi.org/10.1029/2005GB002456, 2006.



Frenger, I., Bianchi, D., Sührenberg, C., Oschlies, A., Dunne, J., Deutsch, C., and et al.: Biogeochemical role of subsurface coherent eddies in the ocean: Tracer cannonballs, hypoxic storms, and microbial stewpots?, Global Biogeochemical Cycles, https://doi.org/doi: 10.1002/2017GB005743, 2018.

Galbraith, E. D. and Martiny, A. C.: A simple nutrient-dependence mechanism for predicting the stoichiometry of marine ecosystems,
Proceedings of the National Academy of Sciences, 112, 8199–8204, https://doi.org/10.1073/pnas.1423917112, 2015.

Gent, P., Willebrand, J., McDougall, T., and McWilliams, J.: Parameterizing Eddy-Induced Tracer Transports in Ocean Circulation Models, Journal of Physical Oceanography, 25, 463–474, https://doi.org/10.1175/1520-0485(1995)025<0463:PEITTI>2.0.CO;2, 1995.

Gordon, A. L., Sprintall, J., Van Aken, H. M., Susanto, D., Wijffels, S., Molcard, R., Ffield, A., Pranowo, W., and Wirasantosa, S.: The Indonesian throughflow during 2004–2006 as observed by the INSTANT program, Dyn. Atmos. Oceans, p. 115–128,
https://doi.org/10.1016/j.dynatnoce.2009.12.002, 2010.

Gruber, N., Hauri, C., Lachkar, Z., Loher, D., Froelicher, T., and Plattner, G.-K.: Rapid Progression of Ocean Acidification in the California Current System, Science, 2012.

Gutknecht, E., Dadou, I., Le Vu, B., Cambon, G., Sudre, J., Garçon, V., Machu, E., Rixen, T., Kock, A., Flohr, A., Paulmier, A., and Lavik, G.: Coupled physical/biogeochemical modeling including $O_2$-dependent processes in the Eastern Boundary Upwelling Systems: application
in the Benguela, Biogeosciences, 10, 3559–3591, https://doi.org/10.5194/bg-10-3559-2013, 2013.

Hallberg, R.: Using a resolution function to regulate parameterizations of oceanic mesoscale eddy effects, Ocean Modelling, 72, 92–103, https://doi.org/https://doi.org/10.1016/j.ocemod.2013.08.007, 2013.

Heinze, C., Maier-Reimer, E., Winguth, A. M. E., and Archer, D.: A global oceanic sediment model for long-term climate studies, Global Biogeochemical Cycles, 13, 221–250, https://doi.org/10.1029/98GB02812, 1999.

Hutchings, L., Pitcher, G., Probyn, T., and Bailey, G.: The chemical and biological consequences of coastal upwelling, in: Upwelling in the Ocean: Modern Processes and Ancient Records, pp. 65–81, John Wiley and Sons Ltd., 1995.

Iida, Y., Takatani, Y., Kojima, A., and Ishii, M.: Global trends of ocean CO2 sink and ocean acidification: an observation-based reconstruction of surface ocean inorganic carbon variables, Journal of Oceanography, https://doi.org/10.1007/s10872-020-00571-5, 2021.

Ilyina, T., Six, K., Segschneider, J., Maier-Reimer, E., Li, H., and Núñez-Riboni, I.: Global ocean biogeochemistry model HAMOCC: Model
architecture and performance as component of the MPI-Earth System Model in different CMIP5 experimental realizations, Journal of Advances in Modeling Earth Systems, https://doi.org/10.1029/2012MS000178, 2013.

Ito, T., Minobe, S., Long, M. C., and Deutsch, C.: Upper ocean O 2 trends: 1958–2015, Geophysical Research Letters, 44, 4214–4223, https://doi.org/10.1002/2017gl073613, 2017.

Johnson, M. T., Liss, P. S., Bell, T. G., Lesworth, T. J., Baker, A. R., Hind, A. J., Jickells, T., Biswas, K. F., Woodward, E., and Gibb, S. W.:
Field observations of the ocean-atmosphere exchange of ammonia: Fundamental importance of temperature as revealed by a comparison of high and low latitudes, Global Biogeochem. Cycles, https://doi.org/doi:10.1029/2007GB003039, 2008.

Jungclaus, J. H., Fischer, N., Haak, H., Lohmann, K., Marotzke, J., Matei, D., Mikolajewicz, U., Notz, D., and Storch, J. S.: Characteristics of the ocean simulations in the Max Planck Institute Ocean Model (MPIOM) the ocean component of the MPI-Earth system model, Journal of Advances in Modeling Earth Systems, 5, 422–446, https://doi.org/10.1002/jame.20023, 2013.

Junker, T., Schmidt, M., and Mohrholz, V.: The relation of wind stress curl and meridional transport in the Benguela upwelling system., J. Mar. Syst., https://doi.org/doi: 10.1016/j.jmarsys.2014.10.006, 2015.





Kalvelage, T., Lavik, G., Lam, P., Contreras, S., Arteaga, L., Löscher, C. R., Oschlies, A., Paulmier, A., Stramma, L., and Kuypers, M. M. M.: Nitrogen cycling driven by organic matter export in the South Pacific oxygen minimum zone, Nature Geoscience, 6, 228–234, https://doi.org/10.1038/ngeo1739, 2013.

Kanzow, T., Cunningham, S. A., Johns, W. E., Hirschi, J. J.-M., Marotzke, J., Baringer, M. O., Meinen, C. S., Chidichimo, M. P., Atkinson, C., Beal, L. M., Bryden, H. L., and Collins, J.: Seasonal Variability of the Atlantic Meridional Overturning Circulation at 26.5°N, Journal of Climate, 23, 5678–5698, https://doi.org/10.1175/2010jcli3389.1, 2010.

Lahajnar, N.: FerryBox measurements at the Namibian upwelling system during the Africana II cruise AFR258, https://doi.org/10.1594/PANGAEA.779132, 2012a.

Lahajnar, N.: FerryBox measurements at the Namibian upwelling system during Maria S. Merain cruise MSM17/3, PANGAEA, https://doi.org/10.1594/PANGAEA.779135, 2012b.

Lahajnar, N.: Nutrients measured with a FerryBox in the Benguela Upwelling System within the Namibian coastal territory during METEOR cruise M103/1 in December 2013 to January 2014, PANGAEA, https://doi.org/10.1594/PANGAEA.854031, 2015a.

Lahajnar, N.: Nutrients measured with a FerryBox in the Benguela Upwelling System during the METEOR cruise M103/2 in January to
February 2014, https://doi.org/10.1594/PANGAEA.854036, 2015b.

Laruelle, G., Cai, W.-J., Hu, X., Gruber, N., Mackenzie, F., and Regnier, P.: Continental shelves as a variable but increasing global sink for atmospheric carbon dioxide, Nature Communications, https://doi.org/doi:10.1038/s41467-017-02738-z, 2018.

Lima, I. D. and Doney, S. C.: A three-dimensional, multinutrient, and size-structured ecosystem model for the North Atlantic, Global Biogeochemical Cycles, 18, n/a–n/a, https://doi.org/10.1029/2003gb002146, 2004.

Liss, P. S.: Gas Transfer: Experiments and Geochemical Implications, in: Air-Sea Exchange of Gases and Particles, pp. 241–298, Springer Netherlands, https://doi.org/10.1007/978-94-009-7169-1_5, 1983.

Maier-Reimer, E.: Geochemical cycles in an ocean general circulation model. Preindustrial tracer distributions, Global Biogeochemical Cycles, 7, 645–677, https://doi.org/10.1029/93gb01355, 1993.

Marcello, F., Wainer, I., and Rodrigues, R. R.: South Atlantic Subtropical Gyre Late Twentieth Century Changes, Journal of Geophysical
Research: Oceans, 123, 5194–5209, https://doi.org/https://doi.org/10.1029/2018JC013815, 2018.

Martin, J. H., Knauer, G. A., Karl, D. M., and Broenkow, W. W.: VERTEX: carbon cycling in the northeast Pacific, Deep Sea Research Part A. Oceanographic Research Papers, 34, 267–285, https://doi.org/https://doi.org/10.1016/0198-0149(87)90086-0, 1987.

Mathis, M., Elizalde, A., and Mikolajewicz, U.: The future regime of Atlantic nutrient supply to the Northwest European Shelf, Journal of Marine Systems, 189, 98 – 115, https://doi.org/https://doi.org/10.1016/j.jmarsys.2018.10.002, 2019.

Mauritsen, T., Bader, J., Becker, T., Behrens, J., Bittner, M., Brokopf, R., and et al.: Developments in the MPI-M Earth System Model version 1.2 (MPI-ESM1.2) and Its Response to Increasing CO2, Journal of Advances in Modeling Earth System, 11, 998–1038, https://doi.org/10.1029/ 2018MS001400, 2019.

Milinski, S., Bader, J., Haak, H., Siongco, A., and Jungclaus, J.: High atmospheric horizontal resolution eliminates the wind-driven coastal warm bias in the southeastern tropical Atlantic, Geophys. Res. Lett., https://doi.org/doi:10.1002/2016GL070530, 2016.

Mohrholz, V., Bartholomaeb, C., van der Plasb, A., and H.U., L.: The seasonal variability of the northern Benguela undercurrent and its relation to the oxygen budget on the shelf, Continental Shelf Research, p. 424–441, 2008.

Monteiro, P. M. S., Dewitte, B., Scranton, M. I., Paulmier, A., and van der Plas, A. K.: The role of open ocean boundary forcing on seasonal to decadal-scale variability and long-term change of natural shelf hypoxia, Environmental Research Letters, 6, 025 002, https://doi.org/10.1088/1748-9326/6/2/025002, 2011.



Najjar, R. G., Sarmiento, J. L., and Toggweiler, J. R.: Downward transport and fate of organic matter in the ocean: Simulations with a general circulation model, Global Biogeochemical Cycles, 6, 45–76, https://doi.org/https://doi.org/10.1029/91GB02718, 1992.

NOAA: National Geophysical Data Center: 2-minute Gridded Global Relief Data (ETOPO2) v2, https://doi.org/10.7289/V5J1012Q, 2006.

Pacanowski, R. C. and Philander, S. G. H.: Parameterization of Vertical Mixing in Numerical Models of Tropical Oceans, Journal of Physical Oceanography, 11, 1443–1451, https://doi.org/10.1175/1520-0485(1981)011<1443:povmin>2.0.co;2, 1981.

Paulmier, A., Kriest, I., and Oschlies, A.: Stoichiometries of remineralisation and denitrification in global biogeochemical ocean models, Biogeosciences, https://doi.org/doi: 10.5194/bg-6-923-2009, 2009.

Paulsen, H., Ilyina, T., Six, K. D., and Stemmler, I.: Incorporating a prognostic representation of marine nitrogen fixers into the global ocean biogeochemical model HAMOCC, Journal of Advances in Modeling Earth Systems, 9, 438–464, https://doi.org/10.1002/2016ms000737, 2017.

Poli, P., Hersbach, H., Dee, D. P., Berrisford, P., Simmons, A. J., Vitart, F., Laloyaux, P., Tan, D. G. H., Peubey, C., Thépaut, J.-N., Trémolet, Y., Hólm, E. V., Bonavita, M., Isaksen, L., and Fisher, M.: ERA-20C: An Atmospheric Reanalysis of the Twentieth Century, Journal of Climate, 29, 4083 – 4097, https://doi.org/10.1175/JCLI-D-15-0556.1, 2016.

Quéré, C. L., Moriarty, R., Andrew, R. M., Canadell, J. G., Sitch, S., Korsbakken, J. I., Friedlingstein, P., Peters, G. P., Andres, R. J., Boden, T. A., Houghton, R. A., House, J. I., Keeling, R. F., Tans, P., Arneth, A., Bakker, D. C. E., Barbero, L., Bopp, L., Chang, J., Chevallier, F.,
Chini, L. P., Ciais, P., Fader, M., Feely, R. A., Gkritzalis, T., Harris, I., Hauck, J., Ilyina, T., Jain, A. K., Kato, E., Kitidis, V., Goldewijk, K. K., Koven, C., Landschützer, P., Lauvset, S. K., Lefèvre, N., Lenton, A., Lima, I. D., Metzl, N., Millero, F., Munro, D. R., Murata, A., Nabel, J. E. M. S., Nakaoka, S., Nojiri, Y., O'Brien, K., Olsen, A., Ono, T., Pérez, F. F., Pfeil, B., Pierrot, D., Poulter, B., Rehder, G., Rödenbeck, C., Saito, S., Schuster, U., Schwinger, J., Séférian, R., Steinhoff, T., Stocker, B. D., Sutton, A. J., Takahashi, T., Tilbrook, B., van der Laan-Luijkx, I. T., van der Werf, G. R., van Heuven, S., Vandemark, D., Viovy, N., Wiltshire, A., Zaehle, S., and Zeng, N.: Global
Carbon Budget 2015, Earth System Science Data, 7, 349–396, https://doi.org/10.5194/essd-7-349-2015, 2015.

Redi, M. H.: Oceanic Isopycnal Mixing by Coordinate Rotation, Journal of Physical Oceanography, 12, 1154–1158, https://doi.org/10.1175/1520-0485(1982)012<1154:oimbcr>2.0.co;2, 1982.

Rixen, T., Cowie, G., Gaye, B., Goes, J., do Rosário Gomes, H., Hood, R. R., Lachkar, Z., Schmidt, H., Segschneider, J., and Singh, A.: Reviews and syntheses: Present, past, and future of the oxygen minimum zone in the northern Indian Ocean, Biogeosciences, 17, 6051–
6080, https://doi.org/10.5194/bg-17-6051-2020, 2020.

Röske, F.: A global heat and freshwater forcing dataset for ocean models, Ocean Modell, https://doi.org/10.1016/j.ocemod.20 04.12.0 05, 2006.

Sabine, C. L., Feely, R. A., Gruber, N., Key, R. M., Lee, K., Bullister, J. L., Wanninkhof, R., Wong, C. S., Wallace, D. W. R., Tilbrook, B., Millero, F. J., Peng, T.-H., Kozyr, A., Ono, T., and Rios, A. F.: The Oceanic Sink for Anthropogenic CO 2, Science, 305, 367–371,
https://doi.org/10.1126/science.1097403, 2004.

Schmidt, M. and Eggert, E.: Oxygen cycling in the northern Benguela Upwelling System: Modelling oxygen sources and sinks, Progress in Oceanography, https://doi.org/doi.org/10.1016/j.pocean.2016.09.004, 2016.

Schmidtko, S., L., S., and M., V.: Decline in global oceanic oxygen content during the past five decades, Nature, https://doi.org/doi:10.1038/nature21399, 2017.

Séférian, R., Berthet, S., Yool, A., Palmiéri, J., Bopp, L., Tagliabue, A., Kwiatkowski, L., Aumont, O., Christian, J., Dunne, J., Gehlen, M., Ilyina, T., John, J. G., Li, H., Long, M. C., Luo, J. Y., Nakano, H., Romanou, A., Schwinger, J., Stock, C., Santana-Falcón, Y., Takano, Y.,





Tjiputra, J., Tsujino, H., Watanabe, M., Wu, T., Wu, F., and Yamamoto, A.: Tracking Improvement in Simulated Marine Biogeochemistry Between CMIP5 and CMIP6, Current Climate Change Reports, 6, 95–119, https://doi.org/10.1007/s40641-020-00160-0, 2020.

Simmons, A. and Gibson, J.: The ERA-40 Project Plan, p. 62, 2000.

Six, K. D., , and Maier-Reimer, E.: Effects of plankton dynamics on seasonal carbon fluxes in an ocean general circulation model, Global Biogeochem. Cycles, p. 559–583, 1996.

Staal, M., te Lintel Hekkert, S., Jan Brummer, G., Veldhuis, M., Sikkens, C., Persijn, S., and Stal, L. J.: Nitrogen fixation along a north-south transect in the eastern Atlantic Ocean, Limnology and Oceanography, 52, 1305–1316, https://doi.org/https://doi.org/10.4319/lo.2007.52.4.1305, 2007.

Steele, M., Morley, R., and Ermold, W.: PHC: A Global Ocean Hydrography with a High-Quality Arctic Ocean, Journal of Climate, 14, 2079–2087, https://doi.org/10.1175/1520-0442(2001)014<2079:pagohw>2.0.co;2, 2001.

Stramma, L., Oschlies, A., and Schmidtko, S.: Mismatch between observed and modeled trends in dissolved upper-ocean oxygen over the last 50 yr, Biogeosciences, https://doi.org/doi:10.5194/bg-9-4045-2012, 2012.

Takahashi, T., Broecker, W., and Langer, S.: Redfield ratio based on chemical data from isopycnal surfaces, J. Geophys. Res., p. 6907–6924, 905 1985.

Takahashi, T., Sutherland, S. C., Wanninkhof, R., Sweeney, C., Feely, R. A., Chipman, D. W., Hales, B., Friederich, G., Chavez, F., Sabine, C., Watson, A., Bakker, D. C., Schuster, U., Metzl, N., Yoshikawa-Inoue, H., Ishii, M., Midorikawa, T., Nojiri, Y., Körtzinger, A., Steinhoff, T., Hoppema, M., Olafsson, J., Arnarson, T. S., Tilbrook, B., Johannessen, T., Olsen, A., Bellerby, R., Wong, C., Delille, B., Bates, N., and de Baar, H. J.: Climatological mean and decadal change in surface ocean pCO2, and net sea–air CO2 flux over the global oceans, Deep 910 Sea Research Part II: Topical Studies in Oceanography, 56, 554–577, https://doi.org/https://doi.org/10.1016/j.dsr2.2008.12.009, 2009.

Tim, N., Zorita, E., Schwarzkopf, F. U., Rühs, S., Emeis, K.-C., and Biastoch, A.: The Impact of Agulhas Leakage on the Central Water Masses in the Benguela Upwelling System From A High-Resolution Ocean Simulation, Journal of Geophysical Research: Oceans, 123, 9416–9428, https://doi.org/10.1029/2018jc014218, 2018.

Vallina, S. and Quéré, C. L.: Preferential uptake of over in marine ecosystem models: A simple and more consistent parameterization, 915 Ecological Modelling, 218, 393–397, https://doi.org/10.1016/j.ecolmodel.2008.06.038, 2008.

Veitch, J., Penven, P., and Shillington, F.: Modeling Equilibrium Dynamics of the Benguela Current System, Journal of Physical Oceanography, 40, 1942–1964, https://doi.org/10.1175/2010JPO4382.1, 2010.

von Storch, J.-S., Haak, H., Hertwig, E., and Fast, I.: Vertical heat and salt fluxes due to resolved and parameterized meso-scale Eddies, Ocean Modelling, 108, 1–19, https://doi.org/10.1016/j.ocemod.2016.10.001, 2016.

Wang, W.-L., Moore, J. K., Martiny, A. C., and Primeau, F. W.: Convergent estimates of marine nitrogen fixation, Nature, 566, 205–211, https://doi.org/10.1038/s41586-019-0911-2, 2019.

Wasmund, N., Struck, U., Hansen, A., Flohr, A., Nausch, G., Gruett-mueller, A., and Voss, M.: Missing nitrogen fixation in the Benguela region, Deep Sea Research Part I: Oceanographic Research Papers, pp. 30–41, 2015.

Williams, R. G. and Follows, M. J.: Physical Transport of Nutrients and the Maintenance of Biological Production, in: Ocean Biogeochem-925 istry, pp. 19–51, Springer Berlin Heidelberg, https://doi.org/10.1007/978-3-642-55844-3_3, 2003.

Woodgate, R. A., Aagaard, K., and Weingartner, T. J.: Interannual changes in the Bering Strait fluxes of volume, heat, and freshwater between 1991 and 2004, Geophys. Res. Lett., https://doi.org/doi:10.1029/2006GL02693, 2006.

Wunsch, C.: The decadal mean ocean circulation and Sverdrup balance, J. Mar. Res., 69, 417–434, 2011.





Zuo, H., Balmaseda, M. A., Tietsche, S., Mogensen, K., and Mayer, M.: The ECMWF operational ensemble reanalysis–analysis system
for ocean and sea ice: a description of the system and assessment, Ocean Science, 15, 779–808, https://doi.org/10.5194/os-15-779-2019,
2019.