# Peer review of "High-resolution modelling of long-term trends in the oxygen and carbon cycles of the Benguela upwelling system"

_Biogeosciences, 2022_

## Author Comment (AC1)

We thank Reviewer #1 for the constructive and detailed review.

In view of the recommendation to restructure the entire paper, we reply only to the major comments. In the following, original comments of reviewer #1 are repeated in italics.

**Scope of the manuscript:** *The title focuses on oxygen and carbon, while the methods focus in great part on a new scheme for nitrogen cycling and two model setups, there are 2 results sections: one focusing on a model comparison mixed with O2 trends and another focusing on pCO2, the introduction mentions mostly just oxygen and acidification. The cycling of carbon is not even mentioned in the introduction. What is the scope of the manuscript?*

Changes in oxygen (deoxygenation) and the carbon cycle (acidification) are emerging problems for coastal areas, such as the Benguela upwelling system (BUS). Mean state and changes result from a complex interplay of physical and biogeochemical processes on very different time scale (meso-scale variability, seasonality, and long-term variability introduced by upwelling of "old" water). Various regional modelling studies put a focus on the mean state and interannual variability of oxygen, but even the mean state of the carbon cycle in the BUS was, to our knowledge, never addressed. The source/sink characteristic of the BUS is an open question.

Our study fills that gap by simulating long-term trends in oxygen and carbon with a model set-up that includes the temporal and spatial variability from meso to decadal scales. The model also includes feedback loops between biological production and OMZ variations due to a comprehensive nitrogen cycle. Nitrogen loss processes also affect alkalinity and could enhance ocean acidification. We show that a poor representation of the OMZ, as found in coarse resolution models, results is a very different answer of the regional anthropogenic carbon uptake.

**Literature:** *The authors miss on many important recent manuscripts that addressed the circulation and oxygen variability in the Benguela upwelling system using similar methods, and many statements read a bit dated. This is very evident in the Introduction, but also in the Results, as many of the findings presented as novel are not such. It also emerges from the fact that the authors recurrently cite the same few papers across the manuscript. The manuscript would really benefit from a better and more up-to-date literature review and a comparison with recent work. I am providing below a list of references that were missed by the authors, although the list is by no mean exhaustive.*

We are grieved that we did not study more carefully the very recent literature. We will include discussions on the latest findings and update the references in the manuscript.

**Paper structure:** *The paper's Introduction is too short and lacks in focus on the topics spanned by the analysis, section 3 "Model results compared to observations" mixes model evaluation with results regarding oxygen trends, and section 4 "Surface pCO2 and decadal trends of the carbon inventory in the BUS" suddenly shifts topic to pCO2. Model evaluation and Results are mixed up, which makes it very difficult to understand what the novel findings are presented by this study. Citations are better suited to the Introduction or Discussion (the latter is currently missing, which leads to this unclear structure). Due to this mix-up, I strongly suggest reorganizing the sections, separating literature review, model evaluation, and the study of the dynamics regulating oxygen distribution and trends. I would almost suggest to split this paper in two dedicated manuscripts: one on O2 and one on pCO2, which would help with both length and clarity. Or else, the connection between the two topics must be strongly clarified in the introduction. The manuscript is currently too long and not well organized in my opinion, the key findings get lost in the text.*

We will follow the suggestion to reorganize the manuscript. As mentioned by the reviewer, the whole model setup and implementation of the extended nitrogen cycle was time-demanding. We agree that large part of theses descriptions could be transferred to the supplement to improve readability.

***A comparison of two models at different resolution:*** *I find many of the results of the model comparison between GR15 and higBUS rather predictable, as it is now expected that a model with lower resolution will not be able, for example, to resolve the upwelling pattern or intensity as well as a model with higher resolution. Many of the plots lack a comparison with the observations. I would suggest the authors to summarize the discussion of the differences between the two models only focusing on the really innovative results, merge the plots and/or move some to a model evaluation section or even a supplement.*

The coarse resolution model is a typical ocean configuration, which is applied in Earth system models for predicting future climate variation.  We also expected most of the results. However, we want to contribute to the ongoing discussion on increasing the complexity of the biogeochemical cycles (e.g. by including an extended nitrogen cycle) in ESMs. Our study shows that, first and foremost, an improved representation of physical variability (i.e. mesoscale resolution) is needed to capture the observed trends in biogeochemistry.

**What is the advantage of resolving the Pacific at such low resolution with the higBUS grid setup?** *Shifted poles grids have been used in several studies before, especially for upwelling systems, mostly using cropped grids that covered only the Atlantic or Pacific basin, depending on the upwelling system of focus. What is the advantage of resolving the entire ocean and especially the Pacific at such low resolution of only 2 to 3 degrees (with all the consequences for atmospheric forcing, representation of currents and biogeochemical cycles) when the focus is the OMZ and the biogeochemistry the BUS?*

All model setups that are restricted to an ocean area/basin need physical and biogeochemical conditions at the boundaries of the model domain. In case of transient simulations, these boundary conditions should ideally include transient signals and their temporal and spatial variabilities. As shown by Espinoza-Morriberon et al. (2021), trends in oxygen within a model domain are sensitive to the variability prescribed at the domain boundaries.

The "beauty" of the global setting applied here is that all tracer distributions and their variabilities within the ocean are the result of a consistent interplay of the physical and biogeochemical processes represented by the model. Potential model deficiencies become visible and are not covered up by artificial boundary conditions. The long spin-up procedure guarantees a quasi-stationary initial condition for the transient simulation. It is also likely that our approach requires less computing time than a two-way nested model configuration.

---

## Author Comment (AC2)

We thank Reviewer #2 for the constructive review.

In view of the recommendation of two reviewers to restructure the entire paper, we only address the major comments. In the following, original comments of reviewer #2 are repeated in *italics*.

The paper focuses on the Benguela upwelling system using a global model and new biogeochemical parameterization scheme forced by 110 years of reanalysis forcing to diagnose deoxygenation and carbon trends in the region. The authors fail to qualify why they focus on the Benguela region. This seems like an odd choice for the questions posed,....

One goal of our study is to show that the representation of meso-scale structures does not only improve the regional biogeochemical mean state, but also affects the simulated long-term trends in an eastern upwelling system. A special focus is on the temporal evolution of the regional oxygen minimum zone. Such a centennial simulation with a global eddy-resolving model is not yet feasible. Therefore, we apply a stretched grid configuration to capture at least the meso-scale activity in the region of interest. Here, the Benguela region is an ideal candidate. By locating the poles on South America and South Africa, we get a high horizontal resolution with a feasible number of grid points due to the relative narrow South Atlantic.

.... and they tread the NBUS and SBUS the same, but they are not the same. The authors need to cite more SBUS papers here – not a lot of representation of the SBUS system on the author team either.

We agree that a more solid evaluation of the SBUS would have been appropriate and it will be provided in a new version of the manuscript. However, we do not understand the statement that we "*tread the NBUS and SBUS the same, but they are not the same*". From a modelling point of view, all processes represented in a biogeochemical model should be valid for any ocean region. None of our parameter settings are dependent on a geographical location. If a process is missing, it should become visible in one of the physical or biogeochemical tracer fields.

Their physical model is not adequate for the research questions. Physical model – agulhas leakage is weak as simulated – something that impacts one of their key points. The Indian Ocean in particular performs poorly in their streamfunction plot on Figure 4. The Agulhas Current is much too weak and too broad in their high resolution model (in fact, it looks better in the low resolution model).

We do not share this opinion of the reviewer at all. We agree that the Agulhas current is rather weak and broad, and that the retroflection is slightly shifted to the south in the high resolution set-up (higBUS). In contrast to the coarse resolution model, however, higBUS produces an Agulhas retroflection where a large bulk of water returns into the Indian Ocean and only a small proportion enters the South Atlantic.

As our focus is on long-term trends in the biogeochemistry, we need a model set-up which is applicable for multi-centennial simulations to achieve a consistent quasi-steady state initial condition. This comes with compromises between model resolution, performance, computer time demands, and available atmospheric forcing fields. E.g. regional models often apply the wind stress correction based on QuikSCAT data which are only available after 1999. Furthermore, transient simulations with two-way nested models are often initialized with spin-up runs of a couple of decades only (all spin-up integrations of the INALT family are 30 years long; Schwarzkopf etal, 2019, https://doi.org/10.5194/gmd-12-3329-2019). This spin-up time is much too short to reduce the model drift at depth and model results will be still close to the initial conditions for most tracers.

When running a model like

this, that includes the Agulhas and the Agulhas retroflection, they need to be explicitly evaluated - the latter by looking at MADT for satellite and model - could be a mean state, and the former by using both satellite and in situ data - the ASCA and ACT data is available on the RSMAS website. While Indian Ocean features, they are highly relevant to the Benguela, especially the southern part, given the importance on water masses, transport of the Benguela, the generation of the shelf-edge jet, turbulence fields in the Cape Basin etc .. Also, there

has been no effort to match their transport of the Agulhas against even published literature (e.g. Beal et al 2015: Capturing the transport variability of a western boundary Jet: results from the Agulhas Current Time-Series Experiment (ACT)). The Agulhas should

be 70-80Sv, the 'mean' Benquela of the order of 20Sv.

We agree that a more solid evaluation of the Agulhas current system would have been beneficial. We will include your suggestions in the updated manuscript.

Their use of a climatological mean in Figure 8, used for an assessment of the coastal upwelling cells, doesn't make sense given that there is a strong seasonal signal in the south and a weak one further north. The southern Benquela upwelling cells will then of course be underestimated here. Given that their focus was on the coastal upwelling here, it is unclear why the authors took vertical velocities at a depth of 100 m when something more like 30/50 m would be more appropriate. Maybe the vertical layers in their model does not allow for this?

As indicated in the manuscript, the vertical grid resolution is 10-12m in the upper 100m. We compared monthly upwelling velocities at 50m and 100m depths and found basically the same variability (seasonal and interannual), although absolute velocities are of course slightly lower at 100m depth. From the biogeochemical point of view, 100m is typically considered the depth of the euphotic layer.

Furthermore, Figure 8 show a climatological mean of the months September to December. The model data presented thus capture the peak upwelling season in the southern BUS well, while slightly underestimating the northern part.

The description of Figure 7 made out that it captured the velocities well, however, their HR model shows intense offshore flow at ~34S which is just not realistic, and their offshore meridional velocities are underestimated, while they are too strong and uniform over the shelf. Instead of separating meridional and zonal

velocities, one possible way forward is to show (as a proxy for surface geostrophic flow): either

MADT (from satellite) compared with mean SSH from the models and/or current speeds overlaid with vectors.

Another metric for evaluation used in this region (given the intense offshore turbulence) is surface eddy kinetic energy (model vs. satellite). This would've given them a good sense of how well the leakage is reproduced.

We will carry out these analyses according to your suggestions.